# Sympathetic neurons can modify the intrinsic structural and functional properties of human pluripotent stem cell-derived cardiomyocytes

Neda Mohammadi[1] (ID), Laura Fedele[2] (ID), Poornaa Chakravarthy[1], Vlad Leonov[1] (ID), Lorenza Tsansizi[1,3], Hui Gu[1], Sama Seyedmousavi[1], Marie-Victoire Cosson[1,3], Andreia S. Bernardo[1,3], Julia Gorelik[1] and Jose L. Sanchez-Alonso[1] (ID)

[1] *Department of Cardiovascular Sciences, National Heart and Lung Institute, Imperial College London, Du Cane Road London, UK*
[2] *Wolfson Sensory, Pain and Regeneration Centre (SPaRC), Guy's Campus, King's College London, London, UK*
[3] *The Francis Crick Institute, London, UK*

Handling Editors: Harold Schultz & T Alexander Quinn

The peer review history is available in the Supporting Information section of this article (https://doi.org/10.1113/JP287569#support-information-section).

**The Journal of Physiology**

**Abstract figure legend** Schematic representation of the key differences between innervated and non-innervated hiPSC-CMs. Innervated hiPSC-CMs (right) exhibit several improved characteristics compared to non-innervated hiPSC-CMs (left). These include enhanced sarcomere organization, a stronger cAMP response under nicotine stimulation, increased contractility with elevated calcium transients and a faster depolarization rate of the action potential.

**Abstract** The sympathetic nervous system densely innervates all cardiac chambers and is a key player in cardiac control, yet this relationship has scarcely been investigated using a stem cell-based model. This study investigates the effects that sympathetic neurons (SNs) have on human pluripotent stem cell-derived cardiomyocytes (hPSC-CMs) *in vitro*, and whether they induce any degree of functional or structural maturity in these conventionally immature cells. SNs were isolated from neonatal rat pups, and cocultured with hPSC-CMs for up to 15 days. Structural changes in hPSC-CMs were analysed by microscopy techniques. Fluorescence resonance energy transfer was used to measure second messenger molecule cAMP production and $\beta$-adrenergic receptor ($\beta$AR) response. Contractile and $Ca^{2+}$ transient activity was measured using CytoCypher. These cocultures promoted hPSC-CM structural elongation and increased sarcomere organization. Furthermore, the $\beta$AR response of cocultured hiPSC-CMs was larger, indicated by increased cAMP production upon neuronal nicotinic stimulation. Faster contraction and ratiometric $Ca^{2+}$ transient peak height and kinetic parameters strongly indicate increased chronotropic response in coculture. Coculture with SNs also elicited an increase in action potential amplitude and depolarization velocity, further confirming that SNs contribute to hiPSC-CM functional maturation. Overall, we have found that SNs modulate hPSC-CMs *in vitro*, inducing a more mature functional response. As an *in vitro* tool, these cocultures could serve as a model of sympathoadrenergic disease, enabling new discovery avenues.

(Received 29 August 2024; accepted after revision 31 January 2025; first published online 26 February 2025)

**Corresponding authors** Julia Gorelik and Jose L. Sanchez-Alonso: Department of Cardiovascular Sciences, National Heart and Lung Institute, Imperial College London, Du Cane Road, London, UK.    Email: j.gorelik@imperial.ac.uk; j.sanchez-alonso-mardones@imperial.ac.uk

**Key points**

- The sympathetic nervous system controls the involuntary 'fight-or-flight' response, with the heart being one of key target organs.
- In certain neuro-cardiac diseases, the input from the sympathetic nervous system is hyper-regulated, and can lead to increased speed or force of the heart's contraction.
- Human induced pluripotent stem cells (hiPSCs) represent a rapidly evolving field which allow us to create a cell of interest and model its structural and functional activity in a dish. Here we have created hiPSC-derived cardiomyocytes (hiPSC-CMs) and cocultured them with sympathetic neurons (SNs).
- We found that SNs are able to modulate structure of the hiPSC-CMs by reducing their circularity and increasing sarcomeric organization, and can significantly increase the speed of contraction and $Ca^{2+}$ handling.
- Together, our data provide a platform to investigate the neuro-cardiac relationship *in vitro*, which could be used for patient-specific disease modelling in future.

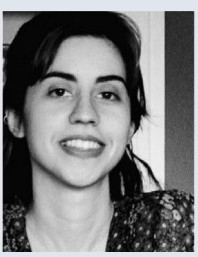

**Neda Mohammadi** obtained her BSc in Pharmacology and Innovative Therapeutics at Queen Mary University of London. She later completed her MSc project at Imperial College London, where she studied the effect of ageing on the maturation of hiPSC-derived cardiomyocytes (hiPSC-CMs). Her PhD project took place in the same department under the supervision of Professor Julia Gorelik; the focus was on the *in vitro* relationship between hiPSC-CMs and sympathetic neurons isolated from neonatal rats to modulate hiPSC-CM structure and function. She now works as a postdoctoral researcher at the University of Oxford investigating all-human neurocardiac models.

# Introduction

The homeostatic relationship between the autonomic nervous system (ANS) and the heart is crucial for normal heart function. The sympathetic branch of the ANS is particularly important as it regulates the body's 'fight or flight' response, a physiological reaction triggered during acute stress. This response is mediated by the ANS, particularly the sympathetic system, through paracrine secretion of neurotransmitters – mainly noradrenaline (NA) – across the synaptic interface, termed the 'neuro-cardiac junction' (NCJ) (Zaglia & Mongillo, 2017), which prepares the organism for rapid action. Abnormalities in the cardiac sympathetic neurons (SNs) contribute to various cardiac diseases and have been directly implicated in their pathogenesis and disease progression (Winbo & Paterson, 2020). Understanding how SNs contribute to cardiomyocyte (CM) development and function could therefore help elucidate the role that innervation of the myocardium plays in both healthy CM development and function within healthy and pathological settings.

The heart is densely innervated during development, with neural projections beginning at the dorsal mesocardium (Hildreth et al., 2009), and later expanding towards the aorta and pulmonary trunk, and eventually the ventricles (Nam et al., 2013). This is mediated via NA released from specialized packages within SNs known as varicosities, displaying a typical 'pearl-necklace' morphology (Prando et al., 2018), which are distributed evenly across the length of multiple neuronal processes (Franzoso et al., 2022). NA stimulates $\beta$-adrenergic receptors ($\beta$ARs) on the surface of CMs, activating a series of downstream signalling cascades that ultimately control cardiac contractility (Franzoso et al., 2016). Thus, unsurprisingly, the absence of functional autonomic innervation during embryogenesis and cardiac development is lethal (Kowalski et al., 2022).

Different coculture of SNs with CMs has been used to model this interaction. Specifically, studies have investigated this intercellular relationship via direct physical contact on coverslips (Oh et al., 2016; Shcherbakova et al., 2007; Winbo et al., 2020), or indirect contact involving the growth of axons through micro-tunnels connecting compartmentalized chips harbouring each cell type (Oiwa et al., 2016; Sakai et al., 2017; Takeuchi et al., 2011). Overall, these studies have demonstrated that CMs have specialized post-junctional-like zones on their surface, where neurites make direct contact with CMs at synaptic varicosities, which is essential for SNs to exert their function on CMs (Franzoso et al., 2016). Fluorescence resonance energy transfer (FRET) analysis has confirmed greater cAMP responses detected in close proximity to these specialized zones (Prando et al., 2018). However, the functional and electrophysiological interaction that takes place at the level of the NCJ *in vitro* is still largely unknown.

The shift towards the use of human pluripotent stem cell-derived (hPSC-derived) alternatives has enabled the use of part-human and all-human models, which have greater physiological relevance, but these have only just started to uncover the functional intricacies downstream of this SN–CM interaction (Bernardin et al., 2022; Kirino et al., 2018; Kowalski et al., 2022; Oh et al., 2016; Winbo et al., 2020). A recent study showed mouse embryo-derived SNs exert a maturation effect on CMs, with cells showing an increase in sarcomere organization and increased connexin-43 expression in coculture (Kowalski et al., 2022). However, these studies focused on few selected maturation parameters. Thus, current understanding of how direct contact of SNs with CMs impacts on their structure and function is limited.

Here we investigate this relationship closely *in vitro*, through the coculture of hPSC-CMs with neonatal rat SNs. Our study assesses both the physical interaction of the cells as well as the role SNs play on several aspects of CM maturity.

We find SNs promote improved structural maturity, faster calcium and contractile dynamics, and increased action potential amplitude with a faster depolarization velocity of CMs. Interestingly, no changes were observed in mitochondrial structure or function. Together these data indicate SNs promote signalling and mechanical maturity but are unable to promote metabolic maturity. These findings contribute to existing research and highlight that SNs play primarily a role in the $\beta$AR/cAMP/Ca$^{2+}$ handling/contraction axis, promoting maturation of this entire pathway and enabling cellular elongation and an increase in sarcomere organization. Overall, our work helps to understand how SNs contribute to cardiac maturation during development. Moreover, it confirms that this coculture model could help understand sympathoadrenergic disease, or how cardiomyocyte-specific diseases could affect the interaction between SNs and CMs, which ultimately could enable the development of new therapeutic treatments in the future.

# Materials and methods

## Ethical approval

We declare that all investigators involved in the study understand the ethical principles under which the journal operates and that the work complies with the animal ethics checklist of the journal (Grundy, 2015).

New-born Sprague–Dawley rat pups (Charles River, Harlow, UK) were killed by cervical dislocation, followed by decapitation as a confirmation method by trained staff, in accordance with the guidelines of Imperial

**Table 1. Summary of antibodies used for flow cytometry**

| Antibody | Volume per $1 \times 10^6$ cells | Source | RRID |
|---|---|---|---|
| SOX2-A488Isotype-A488 | 2.5 µl in 100 µl | BD-Biosciences (561596)BD-Biosciences (557721) | AB_10894382N/A |
| OCT4-PerCP5.5Isotype-PerCP 5.5 | 20 µl in 100 µl | BD-Biosciences (51-900627)BD-Biosciences (51-900627) | N/AN/A |
| NANOG-647Isotype-647 | 5 µl in 100 µl | BD-Biosciences (561300)BD-Biosciences (400130) | N/AAB_396840 |
| SSEA1-PEIsotype-PE | 20 µl in 100 µl | BD-Biosciences (51-9 006268)BD-Biosciences (51-9 006273) | N/AN/A |
| SSEA4-647Isotype-647 | 20 µl in 100 µl | BD-Biosciences (51- 9 006265)BD-Biosciences (51-9006270) | N/AAB_396928 |

College's Animal Welfare and Ethical Review Body (AWERB) and the UK Home Office Animal (Scientific Procedures) Act of 1986. Animal handling was performed by qualified institutional personnel only. The pregnant mothers weighed between 200 and 300 g on the day of mating. They were provided with a standard sniff diet and had free access to water. Housing enrichment included a tunnel and shredded tissue.

### Origin of human pluripotent stem cell lines

The hPSC line IMR90 is an induced pluripotent stem cell line derived from human female fetal lung fibroblasts and was purchased from WiCell Technologies (Madison, WI, USA). Similarly, the hPSC line H9 (WA09, karyotype: 46, female), a human embryonic stem cell line, was also obtained from WiCell Technologies. Since two hPSC lines were used, these will be referred to as hiPSCs for the IMR90 line and hESCs for the H9 line. The hPSC lines were maintained in feeder-free culture conditions on Corning growth factor reduced Matrigel membrane matrix (GFR Matrigel, Corning, New York, NY, USA) for hESCs or geltrex (Thermo Fisher Scientific, Waltham, MA, USA) for hiPSCs, in mTeSR1 maintenance medium (STEMCELL Technologies, Vancouver, Canada) for hESCs or Essential-8 (E8) maintenance medium (Gibco, A1517001) for hiPSCs. Cells were passaged every 4–5 days as aggregates using Gibco Versene solution (Thermo Fisher Scientific) and cell scrapers, at a split ratio of 1:13 (hiPSCs) or 1:8–1:10 (hESCs). All experiments were performed within 20 passages from thawing. All experiments with hESCs were approved by the UK Stem Cell Bank steering committee.

### Flow cytometry

To assess the expression of pluripotency markers in IMR90 stem cells prior to differentiation, flow cytometry was performed. Cells were dissociated using TrypLE (Gibco, 12604013) at 37°C for 5 min, and collected and centrifuged at 300 g for 5 min at room temperature (RT). The supernatant was aspirated and the cell pellet was resuspended in 4% formaldehyde at RT for 15 min. Cells were spun down and resuspended in blocking solution [PBS with 1% bovine serum albumin (BSA), 2% fetal calf serum and 0.5% Tween] at 4°C for 1 h. The sample was passed into a 100 µm mesh to produce single cells, and counted using the EVE counter (Cambridge Bioscience, Cambridge, UK) or NC200 cell counter (Chemometec, Lillerød, Denmark). For staining, the supernatant was removed and cells were resuspended in antibody solution (antibody in PBS with 0.1% BSA and 0.5% Tween per tube), and incubated on ice for 1 h in the dark. Next, cells were washed with 0.1% BSA and 0.5% Tween in PBS, left on an orbital shaker for 5 min and spun down at 300 g for 5 min at 4°C. The wash step was repeated before cells were resuspended in PBS–0.1% BSA, transferred into FACS tubes and analysed using the Flow Cytometry Analyser (Fortessa X20A, Francis Crick Institute, London, UK). Details of the antibodies used are summarized in Table 1.

**PSC-derived cardiomyocyte (hPSC-CM) differentiation.** hPSC differentiation into cardiomyocytes was commenced when cells reached 80–90% confluency, following protocols adapted from Lian et al. (2013) and Maas et al. (2021). These involved the use of small molecules and modulators of the Wnt-signalling pathway. Briefly, cells were treated with 6 or 7 µM CHIR-99021 on day 0 (D0), to account for variations in optimal concentration per differentiation. Cells were topped with RPMI-1640 (Gibco, 11875093) supplemented with 2% B27 minus insulin (Gibco, A1895601) to make RB−, every day until D2. On D3, medium was replaced with 2.5 µM Wnt-C59 (Tocris, 5148, Bristol, UK) or 1 µM IWR1 (Sigma-Aldrich, St Louis, MO, USA) made in RB−. Media were replaced with RB− on D5 and again on D7. On D9, hPSC-CM purification was induced by replacing media with RPMI-1640 minus glucose (Gibco, 11879020) with or without B27 supplement (Gibco, 17504044). Purification lasted a maximum of 4 days before moving the cells to culture medium made with RPMI-1640 and 2% B27 supplement (RB+). For low-yielding differentiations, expansion was made at D11 or D13, by reseeding differentiated hPSC-CMs into new six-well plates and treating with 2 µM CHIR-99021 made in RB+ for 2 days, before replacing with RB+. Culture medium was refreshed with RB+ once every 3 days after D15 until monoculture or coculture preparation.

**Table 2. Primary antibodies used for confocal imaging**

| Primary antibody target | Supplier | Dilution | Host species | Catalogue no. | RRID |
|---|---|---|---|---|---|
| $\alpha$-Actinin | Sigma-Aldrich | 1:500 | Mouse | A7811 | AB_476766 |
| TOH | Sigma-Aldrich | 1:1000 | Rabbit | AB152 | AB_390204 |
| SNAP-25 | Invitrogen | 1:500 | Mouse | MA5-17609 | AB_2789134 |
| CTnT | Abcam | 1:500 | Mouse | ab45932 | N/A |

**Isolation of neonatal rat superior cervical ganglia.** SNs from neonatal rat superior cervical ganglia were isolated following the protocol from Zareen and Greene (2009). Briefly, rat pups (Sprague–Dawley) were killed following cervical dislocation in accordance with ethical guidelines. All the pups from each litter were killed and used in subsequent steps, without distinguishing between males and females. Exclusion criteria were applied in cases of unhealthy litters, defined as instances where more than half of the litter was non-viable. No cell isolation was performed under these circumstances. Superior cervical sympathetic ganglia were isolated and digested into individual SNs following enzymatic digestion with Trypsin 0.25% (Gibco, 25200056) for 30 min at 35°C and mechanical digestion (consecutive passes through glass Pasteur pipettes of increasingly smaller diameters).

**Coculture of hPSC-CMs with SNs.** hPSC-CMs were dissociated between day 25 and 30 of culture and cocultured with SNs onto 7 mm glass-bottomed MatTek dishes (MatTek, P35G-1.5-7-C, Ashland, MA, USA) pre-coated with bovine plasma fibronectin at a ratio of 20:1 (30,000 hPSC-CMs to 1500 SNs per dish), respectively. Cells were fed with RB+, with/without 100 ng/ml of nerve growth factor (NGF) until use. Experiments were performed on hPSC-CMs after 7–15 days in coculture (see figure legends for details). Batch-matched monocultures were seeded in parallel to cocultures following the same protocol and conditions, but without adding the SNs.

**Isolation of neonatal rat ventricular CMs.** Neonatal rat ventricular myocytes (NRVMs) were isolated and used for comparisons in the cAMP response to hiPSC-CMs. The neonatal heart dissociation kit (Miltenyi Biotech, 130-098-373, Cologne, Germany) was used according to the manufacturer's instructions. One kit was used for every five hearts. The hearts were carefully removed and cut into 1 mm$^3$ pieces. Enzyme mixes were prepared as instructed and warmed for 5 min at 37°C. Heart samples were transferred into the C-tubes containing 2.5 ml of enzyme, inverted and incubated at 37°C for 15 min. Cells were incubated and resuspended in 7.5 ml of culture medium and separated with a 70 μm pre-separation filter. The filter was then washed three times in 3 ml of cell culture medium. Cells were centrifuged and resuspended

in 20 ml 10% M199 per 12 hearts in two T75 flasks, and pre-plated in 1% $CO_2$ for 1 h. Medium was collected and cells were seeded at a density of 50,000 cells per 14 mm coverslip.

**Coculture of NRVMs with SNs.** After the isolation of both NRVMs and SNs from the same pups, cells were plated following the same protocol as for hiPSC-CMs, at a 20:1 ratio (50,000 NRVMs to 2500 SNs per dish) and seeded on laminin-coated 14 mm glass-bottomed MatTek dishes (MatTek, P35G-1.5-14-C). Cells were incubated at 37°C, 5% $CO_2$ overnight and fed with MEM medium (Sigma-Aldrich) without the addition of NGF. Half the medium was replaced once every 3 days.

**Immunocytochemistry.** Cells were fixed with 4% paraformaldehyde (PFA) at RT for 15 min, permeabilized with 0.2% Triton X-100 for 10 min, and then incubated for 1 h at RT in filtered blocking buffer (1% BSA and 10% heat-inactivated goat serum in PBS). Cells were treated with primary antibodies diluted in blocking buffer and stored at 4°C overnight. Afterwards, cells were washed three times in PBS, and treated with secondary antibody made in blocking buffer (1:1000, anti-mouse 546, anti-rabbit 488) at RT for 1 h in the dark. Lastly, cells were treated with DAPI diluted 1:10 for 5 min, and kept in PBS until imaging was performed. To assess sarcomere length, CMs were immunostained for Z-discs with $\alpha$-actinin, and F-actin with Alexa Fluor 488 Phalloidin (Invitrogen, A12379, Carlsbad, CA, USA) in blocking buffer. To assess SN neurite growth, tyrosine hydroxylase (TOH) was used as a sympathetic marker. SNAP-25 was used as a synaptic marker. A summary of primary and secondary antibodies is provided in Tables 2 and 3.

**Action potential recordings.** Single hiPSC-CMs and SNs were patched 7–10 days after coculture with time-matched monocultures. Cells were bathed in Tyrode solution (154 mM NaCl, 4 mM KCl, 1 mM $MgCl_2$, 2 mM $CaCl_2$, 5 mM Hepes-NaOH, 5.5 D-glucose; pH 7.4). Amphotericin B (0.22 mM in DMSO) was added to the intracellular solution (125 mM potassium aspartate, 20 mM KCl, 10 mM NaCl, 5 mM $Na_2$-ATP, 10 mM Hepes; pH 7.3) to record action potentials using a Molecular Devices (Sunnyvale, CA, USA) digidata 1440A

**Table 3. Secondary antibodies used for confocal imaging**

| Secondary antibody | Supplier | Dilution | Catalogue no. | RRID |
|---|---|---|---|---|
| Alexa Fluor 546 donkey anti-mouse | Invitrogen Fisher | 1:1000 | A10036 | AB_2534012 |
| Alexa Fluor 488 goat anti-rabbit | Invitrogen | 1:1000 | A11008 | AB_2534069 |
| Alexa Fluor 488 donkey anti-mouse | Invitrogen | 1:1000 | A21202 | N/A |

and a Molecular Devices Multiclamp 700B amplifier at 37°C. Evoked action potentials were recorded using 2 ms current injections (0.5–1.5 nA). Glass borosilicate pipettes (BF100-50-7.5, Sutter Instruments, Novato, CA, USA) were pulled using a P-2000 laser puller (Sutter Instruments) with a pipette resistance of 4–5 MΩ. Signals were digitized at 5 kHz and filtered at 2 kHz with a low pass Bessel filter. The liquid junction potential (LJP) occurs when two liquids in contact with each other have different ion concentrations, and it must be adjusted to accurately measure the cell voltage. LJP was calculated according to the stationary Nernst–Planck equation using LJPcalc (Marino et al., 2014) available at https://swharden.com/software/LJPcalc, and measured LJP as −13.922 mV. Adjusted off-line cells that did not evoke action potentials or with a resting membrane potential above −45mV were excluded for analysis. Action potential recordings were obtained from three independent batches. Analysis (resting membrane potential, peak amplitude, action potential duration APD50 and d$V$/d$t$ max) was performed using the open-source tool BAPTA (Leonov et al., 2023).

**Scanning ion conductance microscopy.** Scanning ion conductance microscopy (SICM) was used to assess morphological changes in the size of a specified 20 × 20 and 7 × 7 μm region of a neurite contact with a CM. A readout of volume, surface area, surface contact and height of the neurite at 2–4, 7–10 or 13–15 days was given. SNs were scanned in hopping mode (Novak et al., 2009). A 100 nm diameter nanopipette was used (80–100 MΩ resistance). The scanning solution (144 mM NaCl, 10 mM Hepes, 1 mM MgCl$_2$ and 5 mM KCl, set to 7.4 pH) was used to fill both the nanopipette and the bath. The nanopipette was made using a P-2000 laser puller (Sutter Instruments). The dish was scanned at 7 × 7 and 20 × 20 μm surface dimensions at RT, and processed using SICMViewer (Novak et al., 2009).

**Confocal imaging and analysis.** To assess the purity of CMs, a 3 × 3 tile scan was produced using a widefield microscope (Zeiss Axio Observer, Oberkochen, Germany). A custom CellProfiler pipeline was used for automated image analysis. The acquired images underwent initial processing to separate the three channels associated with cellular markers. The DAPI channel was used for the identification of nuclei within the

images, while phalloidin was used for the segmentation of individual cells, outlining their morphological boundaries. Subsequently, objects positive for both DAPI and phalloidin were designated as the total cell population. Objects positive for DAPI, phalloidin and α-actinin were classified as CMs and the percentage of CMs was calculated within the total cell population.

A ZEISS LSM-780 inverted confocal microscope was used for all the following imaging. Cell cytoarchitecture was analysed manually according to methods published by Lundy et al. (2013). For hPSC-CM analysis, images were taken as 2 × 2 tile scans with a 20× objective for area and circularity index. For assessment of neurite sprouting, Z-stack images were taken of 5–10 regions in a single coverslip, using the confocal microscope with a 40× objective, with 10 slices and 0.5 μm intervals used between each as a standard setting across all samples. Inclusion criteria required at least one clearly visible neuron cell body (soma) in each image. Images were quantified by saving each as TIFF files, and then SWC files using the NeuTube software for neurite tracing (Feng et al., 2015). Quantification of skeletonized images was performed using the Neuroanatomy SNT plugin on ImageJ (Arshadi et al., 2021).

**Sarcomere analysis.** For the analysis of sarcomere length, organization and alignment, the software tool for automatic quantification (SOTA) was used (Stein et al., 2022), an automatic Python application based on a sarcomere analysis MATLAB script. In brief, single cell images were cut from the original raw confocal images for α-actinin staining. Without any modification, images were uploaded to the SOTA software and automatically analysed. In contrast to other approaches, the method does not require regions of interest (ROIs) to be pre-defined or selected, or images of fluorescent sarcomeres to be aligned manually. SotaTool detects the optimal angle automatically, reducing analysis time and selection bias.

**Transfection of hiPSC-CMs for FRET.** To measure cytoplasmic cAMP levels, the m-Turquoise based [T]Epac[VV] cAMP FRET Epac-SH74 sensor (Klarenbeek et al., 2011), generated in our laboratory, consists of a mutated full-length EPAC1 protein serving as the sensing element. This protein is flanked by mTurquoise (excitation: 434 nm, emission: 474 nm) as the donor fluorophore and a Venus

dimer (excitation: 515 nm, emission: 528 nm) as the acceptor. Upon binding to cAMP, the sensor exhibits a decrease in FRET. Cells were transfected with the plasmid made in Lipofectamine3000 (Thermo Fisher) and serum-free Opti-MEM (Thermo Fisher), according to the manufacturer's instructions. Cultures were incubated at 37°C, 5% $CO_2$ for 48–72 h prior to experiments.

**Real-time cytosolic cAMP measurement.** High-throughput FRET microscopy (MultiFRET) developed in our laboratory was used in combination with a motorized stage (Märzhäuser, Wetzlar, Germany) to select and measure multiple fluorescent transfected hiPSC-CMs. The MultiFRET software (Java plugin) was operated by the Icy software (https://icy.bioimageanalysis.org/) as previously described (Ramuz et al., 2019). In brief, cells were selected using the Nikon TE2000 microscope (Cairn Research, Faversham, UK), and positions were saved in the Micro-Manager. After cell selection, MultiFRET was run and any changes in FRET/fluorescent signals of all selected cells were recorded in real time. Two ROIs were selected per position: the whole cell area visible in the frame and the associated background (no cell area). Images were acquired once every 12 s in the *X*, *Y* and *Z* plane for enhanced focus using an ORCA-Flash 4.0 camera (Hamamatsu Photonics, Japan). After stimulation with 1 and 10 μM nicotine, cells were treated with 100 μM IBMX (a non-specific PDE inhibitor), followed by 10 μM NKH (an adenylyl cyclase activator and stimulator of maximal cAMP response). Drugs were prepared in FRET buffer (10 mM Hepes, 144 mM NaCl, 5 mM KCl, 1 mM $MgCl_2$). Recordings were done in FRET buffer at room temperature. Baseline was measured for ∼10 min until a plateau was reached. Drugs were applied until plateau was reached. All data were calculated as a percentage change from baseline and normalized to the NKH response.

For the ISO dose–response curve, cells were stimulated with ISO concentrations of 0.1, 0.3, 1, 3, 10, 30 and 100 nM, followed by IBMX and NKH (as per nicotine data). The data were plotted as a percentage of the cAMP response, normalized to NKH. ISO concentrations were log-transformed, and the means of each subcolumn were normalized to produce the concentration–response curve. The recorded fluorescence data and automatically calculated FRET ratio shift was exported for further analysis.

The percentage cAMP response normalized to saturator was then calculated as:

$$\% \; cAMP \; response = \frac{FRET \; ratio \; shift \; of \; drug \; response}{FRET \; ratio \; shift \; of \; saturator \; response} \times 100\%$$

**CytoCypher for contraction and $Ca^{2+}$ transient measurement.** The CytoCypher MultiCell system (IonOptix, Westwood, MA, USA) is a high-throughput tool to measure $Ca^{2+}$ and contractile dynamics simultaneously in a single cell (Wright et al., 2020). Here, we use this tool in a novel attempt with hPSC-CMs. Cells were incubated with the ratiometric $Ca^{2+}$ fluorescent dye – Fura-2, AM (Invitrogen, F1221) – prepared in HBSS solution (supplemented with 2 mM $CaCl_2$ and 1 mM $MgCl_2$) with 1% BSA at 37°C, 5% $CO_2$ for 45 min. Afterwards, cells were washed once in HBSS solution without BSA, and incubated at 37°C in 2 ml of HBSS solution per dish for 20 min prior to the experiment. A field stimulator was used to pace the cells at 1 Hz, with 10 ms pulse duration and at 20 V stimulation. Data were processed in CytoSolver V 0.11.2.0 and exported to Microsoft Excel for further analysis. Pixel correlation was used as a measure of contraction in notoriously amorphous hPSC-CMs, relying on Pearson correlation of the movement of pixels in a live frame implemented in the CytoSolver software developed by IonOptix. Thumbnail images of each investigated cell were acquired using a high-speed MyoCam-S3 digital camera, and whole-cell fluorescence photometry of Fura-2 AM was measured with a photomultiplier tube. The region of measurement was automatically selected by the software as the centre of the camera frame, and the full area detected by the photomultiplier was analysed. Fura-2 AM binds to calcium ions in the cytosol, resulting in a shift in its fluorescence excitation spectrum. The CytoCypher system excites the dye at two required wavelengths (340 and 380 nm). The fluorescence intensity ratio (340/380 nm) is then calculated for each cell, providing a ratio proportional to the intracellular calcium concentration.

**Mitochondrial respiration function.** To measure mitochondrial respiration, a Seahorse assay was used with the Seahorse XF kit (Agilent, Didcot, UK). Cells were prepared according to the manufacturer's instructions, first incubated with 10 mM Seahorse XF glucose, 1 mM Seahorse XF pyruvate and 2 mM Seahorse XF

$$FRET \; ratio \; shift = \frac{\left(Average \; baseline \; FRET \; signal \; - \; average \; triggered \; FRET \; response\right)}{Average \; baseline \; FRET \; signal}$$

**Table 4. Human-specific primers designed for RT-qPCR experiments. ZNF384 was used as the housekeeping gene. Primers were designed using primer 3 and crosscheck using primer blast**

| Target gene | Forward sequence | Reverse sequence |
|---|---|---|
| TNNI3 | GCCTCGAGAAAATTGCAGCT | TCATCCACCTTGTCCACACG |
| TNNT2 | GGACTGGAGAGAGGACGAAG | CTGGGCTTTGGTTTGGACTC |
| ACTNT2 | TGGTGTCGGATATTGCTGGT | CATAAGCCCAAGTCTCGTGC |
| JPH2 | GACATCGCGAGAGCTGTG | TTCCTGAAATCTCTGTTTGA |
| MYH7 | CCCACCCAAGTTCGACAAAA | TGTAGATCATCCAGGAGCCG |
| GJA1 | GGAGGGGATAAGAGAGGTGC | ACCACCCGCTCATTCACATA |
| ZNF384 | CAGGACTGATGACTGCTGGA | AGACAATCATGGGAGCCGAA |

L-glutamine. Then, the Cell Mito Stress Kit was used to assess the mitochondrial respiration function of the cells. First, oxygen consumption rate (OCR) was measured as the baseline of the mitochondrial respiration. Next, using the Seahorse Xfe96 Analyzer injector (Agilent), cells were exposed to 2.5 mM oligomycin, 2 mM FCCP (carbonyl cyanide *p*-trifluoro-methoxyphenyl hydrazone) and 1.08 mM rotenone/antimycin in subsequent steps. After the assay, cells were washed with PBS once and fixed with 4% PFA for 15 min. After fixation, cells were washed with PBS twice and incubated with DAPI (1:10,000) in PBS for 10 min to count nuclei per well, to which data were normalized. This allowed the calculation of the basal respiration, ATP-linked respiration and maximal respiratory capacity, of the cells in the tested media.

**Mitochondrial potential assessment.** Once ready, cells were dissociated with 450 ml dissociation medium per well at 37°C for 12 min. Next, cells were topped up with 450 ml support media with Rock inhibitor (Tocris, 1254) and divided into seven tubes. Cells were spun down, and the cell pellet resuspended in unstained solution, single staining solutions or all staining solutions combined. Unstained control solution consisted of support media supplemented with Rock inhibitor. Other staining solutions included MitoProbe™ TMRM Assay Kit solution (1:1000) (Thermo Scientific, M20036), CD36 antibody solution (1:25) (Biolegend, 336205, San Diego, CA, USA), MitoTracker™ Red CMXRos solution (1:50,000) (Invitrogen, M46752) or all of the above; all staining solutions were made in support media supplemented with Rock inhibitor. Cells were incubated in staining solutions for 30 min at 37°C, 5% $CO_2$. Afterwards, cells were spun down and resuspended in DAPI (1:10,000) in PBS. Lastly, cells were transferred to FACS tubes and analysed using a flow cytometry analyser (Fortessa X20A, Francis Crick Institute).

**Mitochondrial morphology assessment.** To assess mitochondrial function, hPSC-CMs were seeded at a lower density of 10,000 cells per dish to identify individual cells, and stained with the red fluorescent mitochondrial dye MitoTracker CMXRos, which accumulates with increasing membrane potential. Medium was aspirated from the dish and replaced with 100 nM MitoTracker CMXRos made in RPMI-1640, prewarmed to 37°C. Cells were treated with 2 ml of the dye preparation and incubated for 1 h at 37°C. Afterwards, the dye was aspirated, and cells were washed in DPBS prewarmed to 37°C twice before being fixed in 4% PFA for 15 min at RT. Cells were subsequently stained for the cardiac marker $\alpha$-actinin, and nuclei with DAPI. Images were taken in a single plane, and analysed using the mitochondrial network analysis macros (MiNA) using an ImageJ plugin (Valente et al., 2017). This is a semi-automated method of analysis which binarizes the image to produce a morphological skeleton from which specific parameters including the mitochondrial footprint and branch length mean were calculated. Images were split into independent channels and hPSC-CMs were identified from non-CMs as indicated by $\alpha$-actinin expression.

**Two-step RT-qPCR.** RNA was extracted using the RNeasy Plus Mini Kit (Qiagen, Valencia, CA, USA) in accordance with the manufacturer's instructions. The extraction was performed on monocultures and cocultures, both treated with 100 ng/ml NGF and cultured for 7–10 days following standard coculture procedures. cDNA synthesis was performed using the GoScript Reverse Transcription System (Promega, Madison, WI, USA). Real-time quantitative PCR (RT-qPCR) was performed following amplification with the QuantiNova SYBR Green PCR kit (Qiagen) on the Applied Biosystems QuantStudio 6 Flex Real-Time PCR system. All equipment and reagents used, apart from those provided in the kits, were RNase-free. Primers were designed specifically for human genes to measure gene expression in the hiPSC-CMs without cross-detecting mRNA from rat SNs (Table 4).

**Statistical analysis.** Statistical analysis is reported for each experimental section. Normality of the data distribution was assessed with the Shapiro–Wilk test.

Generally, unpaired non-parametric analysis was used in GraphPad Prism V 8.4.3, including the Mann–Whitney test two-group comparison or Kruskal–Wallis test for multiple group comparisons. Statistical significance is defined as $P \leq 0.05$, and data are presented as mean $\pm$ SD.

## Results

### SNs connect to hiPSC-CMs and respond to nicotine within *in vitro* cocultures

To study the role of the NCJ in CM function and structure, we first confirmed the pluripotency of IMR90 stem cells with flow cytometry (Fig. 1*A*). The successful generation of hiPSC-CMs was next confirmed by staining mono-cultures with $\alpha$-actinin and quantifying the amount of hiPSC-CMs in the total population of cells (Fig. 1*B*, *C*). Next, we focused on establishing coculture conditions between hiPSC-CMs and rat SNs. We determined that exogenous NGF is needed for appropriate sprouting of the neurites in a time suitable for our experiments, as the cocultures without the addition of NGF showed a very limited neuronal sprouting after 15 days (Fig. 2*A*, *B*). A significant increase was observed in the number of neurite branch points ($P < 0.001$; $P = 0.01$), neurite path length ($P < 0.001$) and neurite surface area ($P < 0.001$) in cocultures over time (from 2 to 15 days) (Fig. 2*C*). Of note, we saw neurites displaying typical 'pearl-necklace' morphology, possibly suggesting that SNs contain varicosities and were contacting the hiPSC-CMs (Fig. 2*D*, white arrows). In addition to time in coculture, the presence of NGF significantly affected neurite sprouting with respect to the number of branch points ($P = 0.00440$, $P < 0.001$, $P = 0.0429$), neurite path length ($P < 0.001$) and surface area ($P < 0.001$) (Fig. 2*E*).

Next, to identify synaptic activity in cocultures, cells were stained for SNAP-25, a member of the presynaptic SNARE complex, and essential component of synaptic activity in neurons (Fig. 3*A*). To confirm the functional activity of SNs, they were stimulated with 10 μM nicotine (followed by a washout period) and recorded under whole-cell configuration. Nicotine treatment elicited action potentials on SNs and confirmed the use of nicotine for further experiments (Fig. 3*B*). In parallel, to investigate the morphological changes in neurite size over time and determine if this could have an impact on the size of the NCJ, topographical analysis by SICM was performed. The NCJ is defined as the area of contact between the SN neurites and the CM surface. A schematic illustrating the setup and the parameters used to assess neurite surface topography is shown in Fig. 3*C*. Representative examples of 3D topographical scans are shown in Fig. 3*D*. Analysis of the 3D topographical scans found no significant change in neurite size with respect to volume ($P > 0.999$), surface area ($P > 0.999$), surface contact ($P > 0.999$) or

height ($P > 0.999$, $P = 0.793$) over time (Fig. 3*E*). No differences in morphology were observed in any of the groups by SICM. A period of 7–10 days was sufficient to allow for neurite growth and expansion without the neuronal network saturating the field of view upon NGF supplementation.

Together, these data show that cocultures can be successfully achieved after only a few days when in the presence of NGF, and that SNs within these cultures express functional synaptic proteins and respond to external stimuli, suggesting that SNs and CMs could form a functional NCJ. Importantly, 7–10 days in coculture was chosen for all the experiments, as our data from Fig. 2*C* and *E* showed our immunocytochemistry data supported the formation of the NCJ at this time point.

### SNs promote cell elongation and structural maturation of the CMs, but do not contribute to metabolic changes or gene expression

Stronger contraction normally elicits sarcomeric changes leading to metabolic, cellular and structural adaptations which enable cells to contend with the contraction demands. Thus, we next performed a detailed cytoarchitecture and metabolic function analysis of hPSC-CMs grown in the presence or absence of SNs. To assess the SN-induced structural changes in hiPSC-CMs, we analysed cell shape following immunostaining (Fig. 4*A*). Results showed no significant change in cell area (Fig. 4*B*), but the circularity index was significantly reduced in cocultures ($P < 0.001$; Fig. 4*C*), suggesting that SNs induce elongation of hiPSC-CMs. Sarcomere analysis of the $\alpha$-actinin staining using SOTA automatic analysis software (Stein et al., 2022) showed that cocultures present a statistically significant reduction in sarcomere length ($P = 0.00880$), and a significant increase in sarcomere organization ($P = 0.0174$), but no significant change in their alignment index (Fig. 4*D*). The same experiments were performed on the embryonic H9 stem cell line (hESC-CMs). In the presence of SNs, hESC-CMs were also elongated (Fig. 4*E*, *F*), their sarcomere length was reduced ($P = 0.0467$) and their sarcomere organization ($P = 0.0360$) was increased (Fig. 4*H*).

Next, we determined if SNs affected mitochondrial abundance, shape and organization. Live staining of mitochondrial networks in hPSC-CMs was analysed (Fig. 5*A*), showing no significant difference in mitochondrial footprint ($P = 0.902$, Fig. 5*B*) or the expansion of the mitochondrial network, as measured through mean branch length ($P = 0.102$; Fig. 5*B*) on hiPSC-CMs. The same results were found on hESC-CMs in the mitochondrial footprint ($P = 0.271$, Fig. 5*C*), with no significant difference found in mean branch length ($P = 0.138$, Fig. 5*C*). These data were further

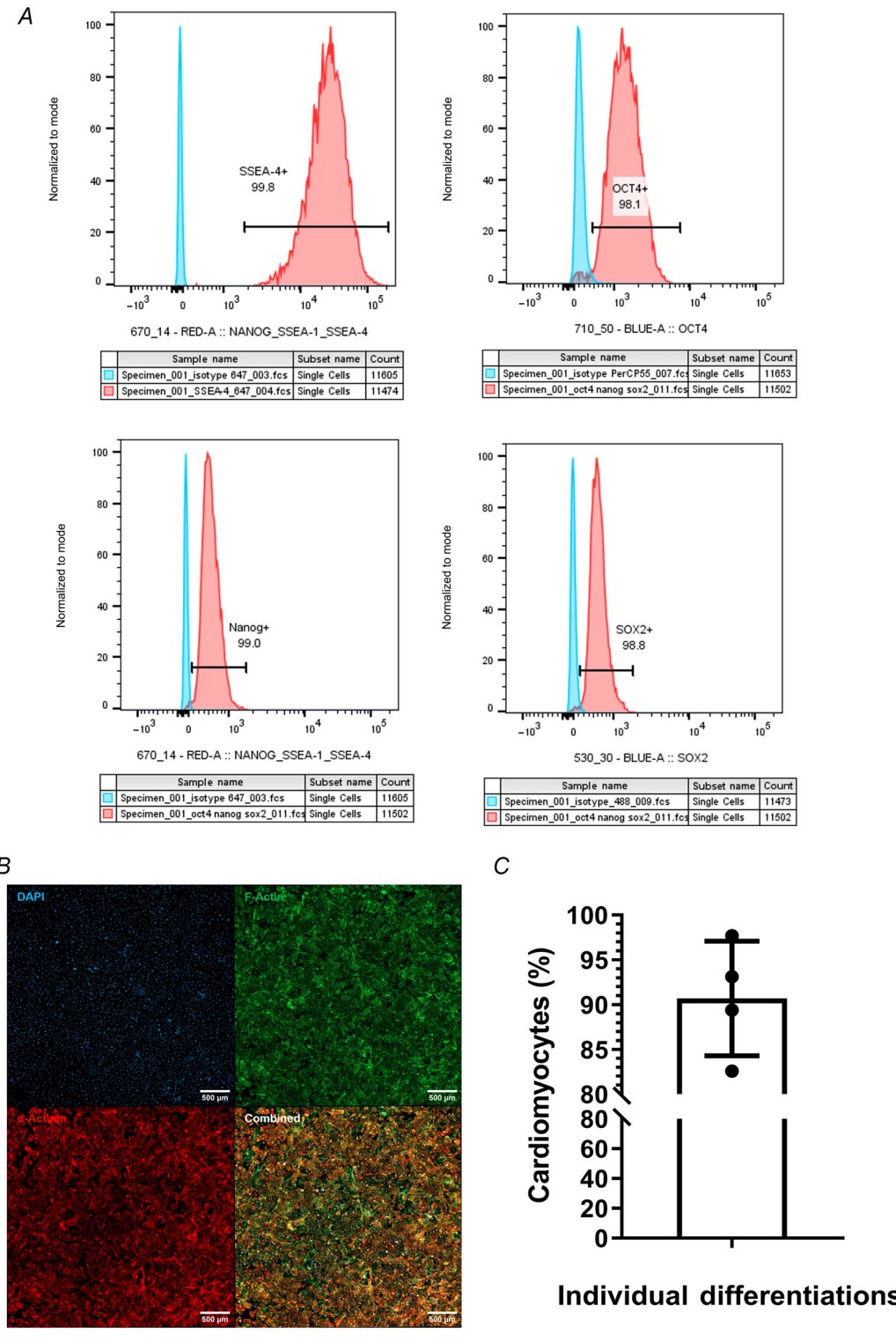

**Figure 1. Characterization of hiPSC-CMs differentiated from stem cells**
Assessment of the efficiency of differentiation into hiPSC-CMs. *A*, flow cytometry plots showing expression of key pluripotency markers in IMR90 iPSCs, including SSEA-4, OCT4, Nanog and SOX2. *B*, widefield microscopy was used to take a 3 × 3 tile scan of monocultures, staining for nuclei with DAPI (blue), F-actin with phalloidin (green) and the key CM marker α-actinin (red). Scale bar = 500 μm. *C*, the proportion of α-actinin-positive cells was calculated as a percentage of α-actinin-positive cells within the total number of cells in immunostained cultures as in *B*. An average of 90.69% α-actinin-positive cells was observed. *N* = 4 differentiations. [Colour figure can be viewed at wileyonlinelibrary.com]

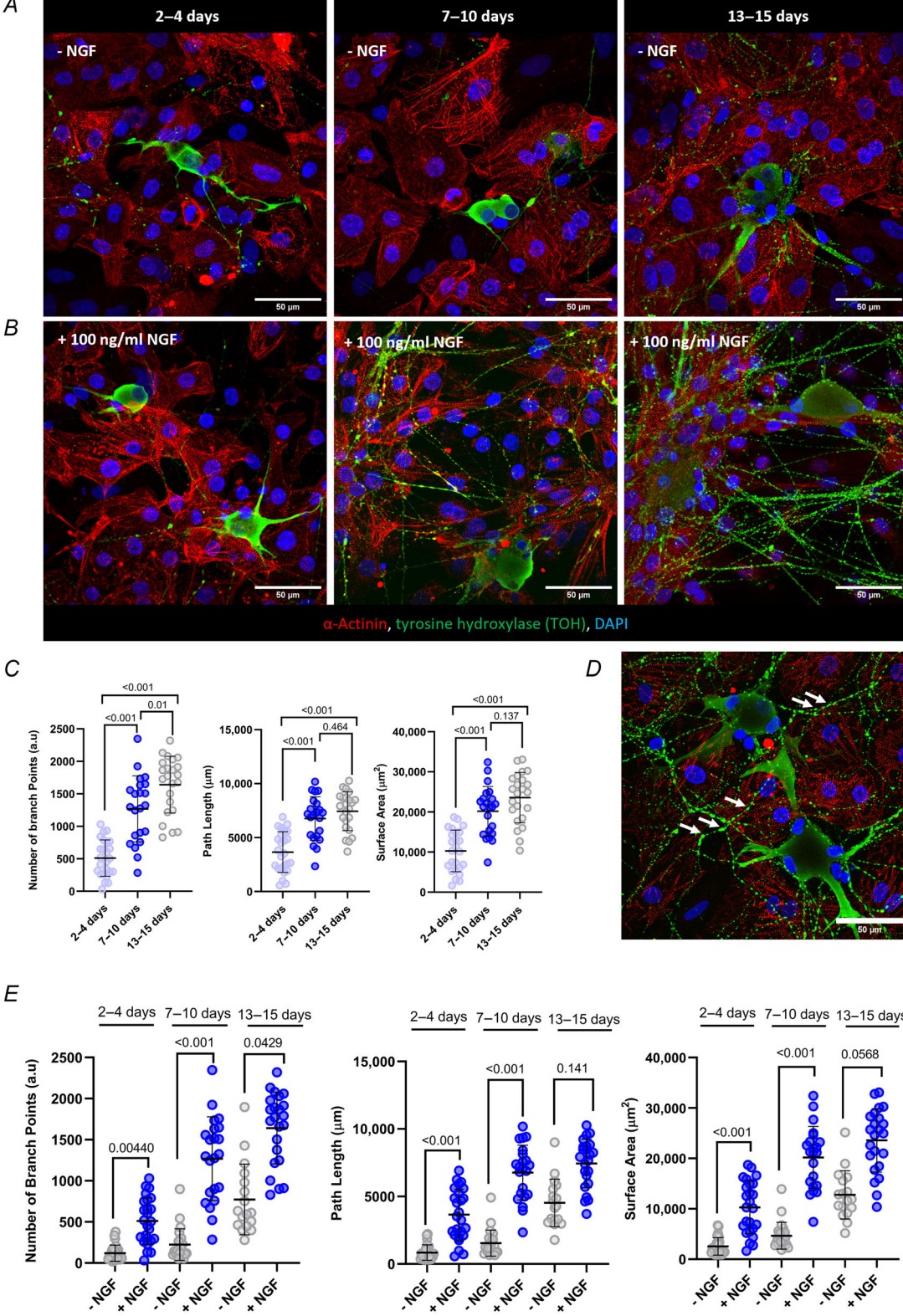

**Figure 2. Exogenous NGF promotes the expansion of SN neurites**

Cocultures were maintained in standard medium or treated with 100 ng/ml NGF. Representative images of hiPSC-CMs and SNs cocultured at three time points stained for α-actinin (red), tyrosine hydroxylase (TOH, green) and nuclei with DAPI (blue): *A*, without NGF; or *B*, with NGF. Maximum projection of Z-stacks with a 40× objective. Scale bar = 50 μm. *C*, significant increase in the number of branch points (*P* < 0.001), path length (*P* < 0.001)

and surface area ($P < 0.001$) of neurites over time in coculture with NGF between 2–4 days ($N/n = 4/25$) and 7–10 days ($N/n = 4/22$), and 2–4 and 13–15 days ($N/n = 4/23$). Data presented as mean with SD, one-way ANOVA. D, representative image of cocultures stained for TOH, with arrows indicating varicosities with a 40× objective, and up to 15 days in coculture with NGF. Scale bar = 50 μm. E, comparisons between supplementation with or without NGF show a significant increase in the number of branch points ($P = 0.00440$, $P < 0.001$), path length ($P < 0.001$) and surface area ($P < 0.001$) at 2–4 and 7–10 days. There was also a significant increase in the number of branch points between 7–10 and 13–15 days of coculture ($P = 0.0429$). Images of cocultures were taken at 2–4 days (NGF $N/n = 4/25$, without NGF $N/n = 4/24$), at 7–10 days (NGF $N/n = 4/22$, without NGF $N/n = 4/23$) and 13–15 days (with NGF = 4/23, without NGF = 3/17). Data presented as mean with SD, Kruskal–Wallis test. $N/n$ = biological replicate/cell number. [Colour figure can be viewed at wileyonlinelibrary.com]

confirmed with flow cytometry; NGF did not significantly affect mitochondrial membrane potential ($P = 0.651$) or coculture ($P = 0.942$, Fig. 5D), or the abundance of mitochondria in hiPSC-CMs in monoculture ($P = 0.830$) or coculture ($P = 0.948$, Fig. 5E). Seahorse analysis found no significant difference in the OCR of monocultures or cocultures, in either hiPSC-CMs (Fig. 6A, B) or hESC-CMs (Fig. 6C, D), suggesting that SNs do not change the metabolic activity of hPSC-CMs.

To investigate whether SNs contribute to changes in gene expression underlying the observed effects, we assessed mRNA expression levels of JPH2, ACTN2, TNNI3, TNNT2, MYH7 and GJA1 using RT-qPCR (Fig. 6E). These genes encode junctophilin, alpha-actinin 2, cardiac troponin I, cardiac troponin T, $\beta$-myosin and gap junction protein alpha, respectively. No significant changes were observed in expression levels of these genes on hiPSC-CMs, suggesting that the maturation of structural proteins results from functional changes in the cells rather than alterations in gene expression.

## The NCJ present in cocultures is functional and can regulate the cAMP pathway within CMs

To ascertain whether SN stimulation induces a response within hiPSC-CMs, we stimulated cocultures with nicotine and measured the cytosolic cAMP response using the mTurquoise-EPAC-$^{cp173}$Venus-Venus ($^{T}$Epac$^{VV}$) FRET sensor (Klarenbeek et al., 2011). Representative traces for nicotinic stimulation show a stepwise response to $\beta$AR stimulation (Fig. 7A). As expected from a functional NCJ, SN stimulation elicited a cAMP response in cocultured hiPSC-CMs, which was stronger with increasing concentrations of nicotine ($P < 0.001$). By contrast, nicotine induced a minimum cAMP response in monocultures ($P = 0.00930$ between 1 and 10 μM nicotine; $P < 0.001$), confirming the cAMP response seen in cocultured hiPSC-CMs was attributable to stimulation of SNs (Fig. 7B). Responses to individual concentrations of nicotine were also higher in coculture than monoculture ($P < 0.001$; Fig. 7C). Interestingly, IBMX, a non-selective phosphodiesterase (PDE) inhibitor which raises intracellular cAMP, induced a stronger cAMP response in cocultures than in monocultures ($P < 0.001$), suggesting

SN innervation could play a role in the PDE balance (Fig. 7C). A similar trend showing increased cAMP response in coculture was observed when comparing the response to 0.1 and 0.3 μM forskolin (a prominent adenylyl cyclase activator) in hiPSC-CMs (Li et al., 2023).

Next, we investigated whether SNs have an effect on $\beta$AR response. Thus, we stimulated the hiPSC-CMs with the $\beta$ARs agonist isoproterenol (ISO), and measured the cAMP response only in CMs (Bardsley et al., 2018). The $EC_{50}$ of ISO was not significantly different between monocultures and cocultures (Fig. 7D), suggesting that the expression and balance of $\beta$AR is not affected by SNs. We also tested the effect of ISO on cocultures of SNs with NVRMs, which showed the same results (Fig. 7D), therefore confirming that coculture of SNs with CMs (rat or hiPSC-derived) does not affect $\beta$AR responses on the CMs regardless of the species. In summary, we show that the NCJ is functional and can trigger a $\beta$AR response on the innervated CMs, but it does not seem to affect whole-cell $\beta$AR expression or the response to ISO.

## SNs promote improved Ca$^{2+}$ and contractile function within hPSC-CMs

Given the results observed, we were interested to see whether SNs contribute to a more mature hPSC-CM functional activity, by determining whether cocultures displayed improved contraction and Ca$^{2+}$ handling. First, we measured the beats per minute (bpm) in monocultures and cocultures to study whether SNs affected the spontaneous beating of CMs. hiPSC-CMs monocultures showed an average of 45 bpm (SD = 15.17), compared to 48 bpm (SD = 25.09) for cocultures (measured from six batches). Similarly, hESC-CMs showed 38 bpm (SD = 12.92) in monocultures and 42 bpm (SD = 10.92) in cocultures (measured from four batches). The slight increase observed in cocultures was not statistically significant in either case. Given the relatively high variability in bpm between batches and the influence of the beating rate on calcium transient parameters, all subsequent experiments were conducted under electrical pacing conditions. For analysis of Ca$^{2+}$ dynamics, we pre-incubated cells with the ratiometric Ca$^{2+}$ dye Fura-2 AM. Then, we assessed cells at baseline for their

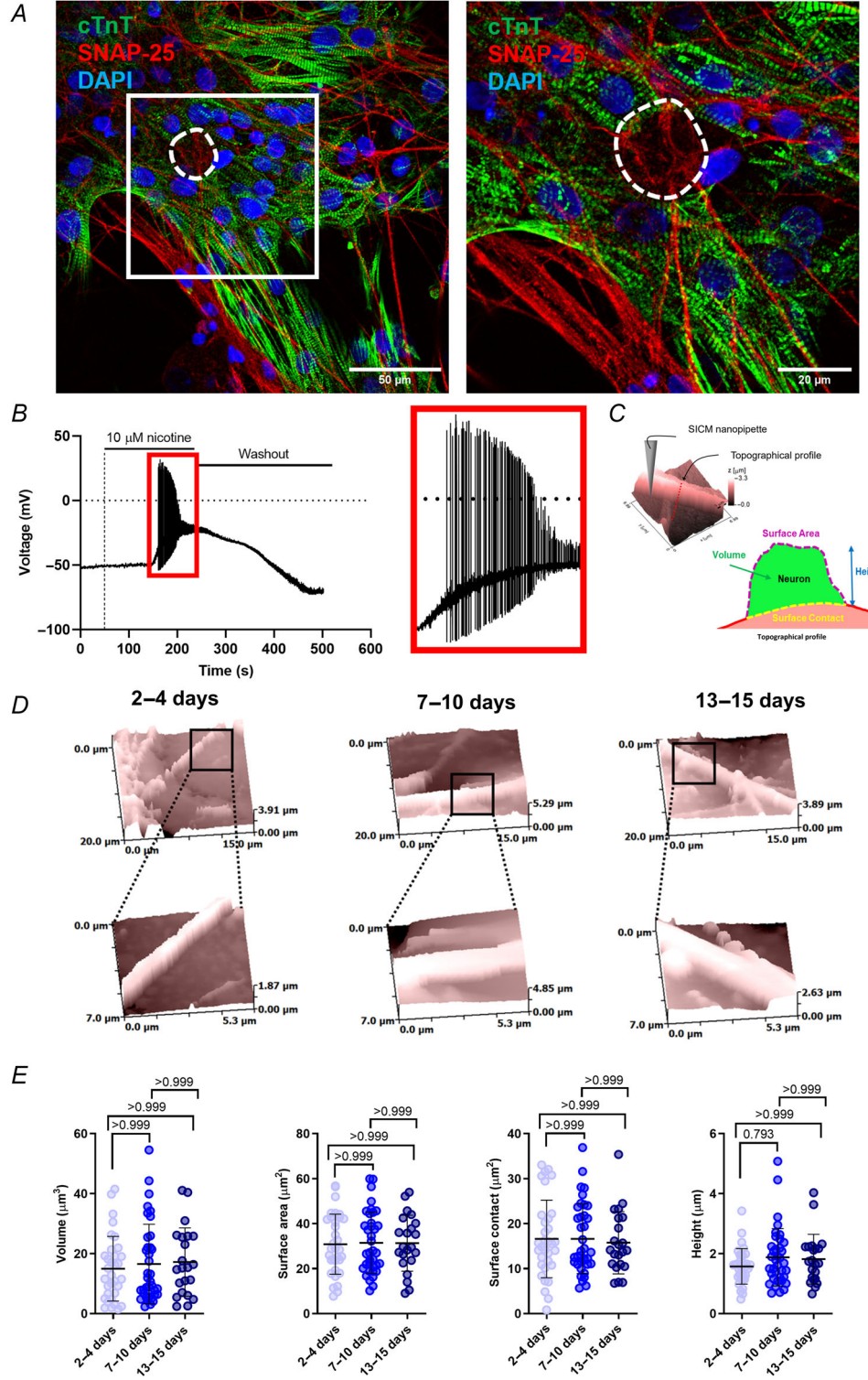

**Figure 3. Functional activity of SNs and morphological characteristics of the neurocardiac junction (NCJ)**
Functional activity of SNs at sites of contact was investigated with respect to the expression of a select synaptic protein and action potential firing in response to nicotinic stimulation. *A*, Left: representative immunofluorescence image of cocultured hiPSC-CMs with 100 ng/ml NGF, stained for a prominent marker of neuroexocytosis, SNAP-25 (red), along the neurites of SNs. hiPSC-CMs were stained for cTnT (green), and nuclei stained with DAPI (blue). Right: a 2× zoom showing where the neuronal soma (dashed line) is located, from an above *Z*-stack image. Scale bars = 50 µm, 20 µm. *B*, whole-cell current clamp recording of a single SN at rest and upon exposure to 10 µM

nicotine. *C*, schematic illustrating the setup for the scanning ion conductance microscope (SICM) to record changes in the NCJ topography over time. *D*, representative SICM measuring topographical images of 20 × 20 μm (top) and higher resolution 7 × 7 μm scans (bottom) of SNs at the NCJ at three stages in coculture, with no significant difference. *E*, summary of the volume ($P > 0.999$), surface area ($P > 0.999$), surface contact ($P > 0.999$) and height ($P > 0.999$, $P = 0.793$) after 2–4 days ($N/n = 5/33$), 7–10 days ($N/n = 5/37$) or 13–15 days ($N/n = 4/22$) in coculture with hiPSC-CMs. No significant difference was detected between each stage in culture. Data presented as mean with SD, Kruskal–Wallis test. $N/n$ = biological replicate/cell number. [Colour figure can be viewed at wileyonlinelibrary.com]

contraction by measuring pixel correlation, and for their $Ca^{2+}$ transients under paced conditions at 1 Hz and 37°C. Representative traces are shown (Fig. 8*A*, *B*). In cocultures, hiPSC-CMs displayed significantly larger contraction amplitude, measured as pixel correlation peak height ($P < 0.001$; Fig. 8*C*), and faster departure ($P = 0.0160$; Fig. 8*D*) and return velocities ($P = 0.00120$; Fig. 8*E*). This demonstrates that innervated hiPSC-CMs can contract more and with faster kinetics. For the $Ca^{2+}$ transients, cocultured hiPSC-CMs displayed larger a $Ca^{2+}$ transient amplitude trend (peak height), but without statistical significance ($P = 0.123$; Fig. 8*F*). Departure velocity was indeed faster ($P < 0.001$; Fig. 8*G*). However, there was no significant change in return velocity ($P = 0.359$; Fig. 8*H*). A set of experiments was also reproduced in hESC-CMs, showing similar significant differences and trends as the hiPSC-CMs (Fig. 9).

Next, we applied nicotine in the bath solution at concentrations of 1 and 10 μM to the monoculture (Fig. 10*A*) and cocultures (Fig. 10*B*). We first noted that hiPSC-CMs in monoculture showed a significant decrease from baseline in pixel correlation peak height ($P < 0.001$) after nicotine (Fig. 10*C*), but with no significant difference between both applications ($P > 0.999$). Similar results were observed in the contraction kinetics for both departure ($P < 0.001$, $P = 0.0502$, $P = 0.0861$; Fig. 10*D*) and return velocities ($P = 0.0861$, $P = 0.00499$, $P > 0.999$; Fig. 10*E*). Unfortunately, high-affinity calcium dyes have been reported to buffer intracellular calcium and alter physiological behaviour (Fast, 2005; Wokosin et al., 2004). This limitation is probably exacerbated under the conditions of our recordings – constant pacing at 1 Hz, 37°C, and Fura-2 AM loading – which may cause a progressive decrease in the contraction capacity of the cells over time. As a result, these effects could be masking the positive inotropic effects that would otherwise be expected in the cocultures. Considering this, hiPSC-CMs in coculture are positively affected by the application of nicotine, as their peak height recovers after 10 μM nicotine application ($P = 0.125$), following a drop from baseline after 1 μM nicotine ($P = 0.00959$; Fig. 10*F*). There was no significant difference in peak height response between both nicotine applications ($P > 0.999$; Fig. 10*F*). The speed of contraction remained constant ($P = 0.102$, $P = 0.760$) except for a significant increase in departure velocity after 10 μM nicotine ($P = 0.00330$; Fig. 10*G*). In contrast, there was no significant difference observed in return velocity

after nicotine ($P = 0.525$, $P > 0.999$, $P = 0.248$; Fig. 10*H*). Similar results were observed when hESC-CMs were used. Contraction parameters in monoculture dishes showed a decline over time, while cocultures exhibited a positive effect from nicotine that counteracted the decay observed in monocultures (Fig. 11).

For $Ca^{2+}$ transients, representative traces are shown (Fig. 12*A*, *B*). There was a rundown of the $Ca^{2+}$ transient peak height from baseline over time in monocultures ($P = 0.00939$, $P < 0.001$; Fig. 12*C*). Departure velocity was not significantly changed ($P = 0.0825$) except for a reduction after 10 μM nicotine ($P < 0.001$), with no significant difference between both nicotine applications ($P = 0.179$; Fig. 12*D*). There was no significant change in the $Ca^{2+}$ transient return velocity in monoculture ($P = 0.591$, $P = 0.440$, $P > 0.999$; Fig. 12*E*). In contrast, we observed a stabilization of the $Ca^{2+}$ transient peak height in cocultures under nicotinic stimulation ($P = 0.132$, $P = 0.322$, $P > 0.999$; Fig. 12*F*), similar to the pixel correlation data. However, the $Ca^{2+}$ transient kinetics were not significantly changed in departure velocity ($P > 0.999$; Fig. 12*G*) or return velocity ($P = 0.360$, $P = 0.925$, $P > 0.999$; Fig. 12*H*). Similar results were observed in hESC-CMs (Fig. 13), except for departure velocity (Fig. 13*G*), where nicotine significantly increased departure velocity in hESC-CM cocultures ($P < 0.001$).

Finally, we performed action potential (AP) recordings of the hiPSC-CMs in monoculture and coculture. A representative AP trace of each group is shown (Fig. 14*A*). Patch-clamp recordings of APs were performed in monocultures and cocultures, confirming a significantly higher amplitude of the AP ($P = 0.006$) and faster depolarization rates ($P = 0.0418$) in cocultures, without significant changes in resting membrane potential ($P = 0.0614$) or repolarization rate, indicated by AP duration 90 (APD90) ($P = 0.119$; Fig. 14*B*). We also characterized the proportion of ventricular, atrial and nodal-like hiPSC-CMs based on their APD90/APD50 ratio (Matsa et al., 2011). The analysis revealed that in monocultures, the distribution was 44% ventricular-like, 24% atrial-like and 32% nodal-like hiPSC-CMs. Similarly, in cocultures, the distribution was 48% ventricular-like, 26% atrial-like and 26% nodal-like hiPSC-CMs. These findings align with previous reports utilizing standard iPSC-derived cardiomyocyte differentiation protocols (Dark et al., 2023), and suggest that SNs had minimal impact on the population distribution of hiPSC-CMs.

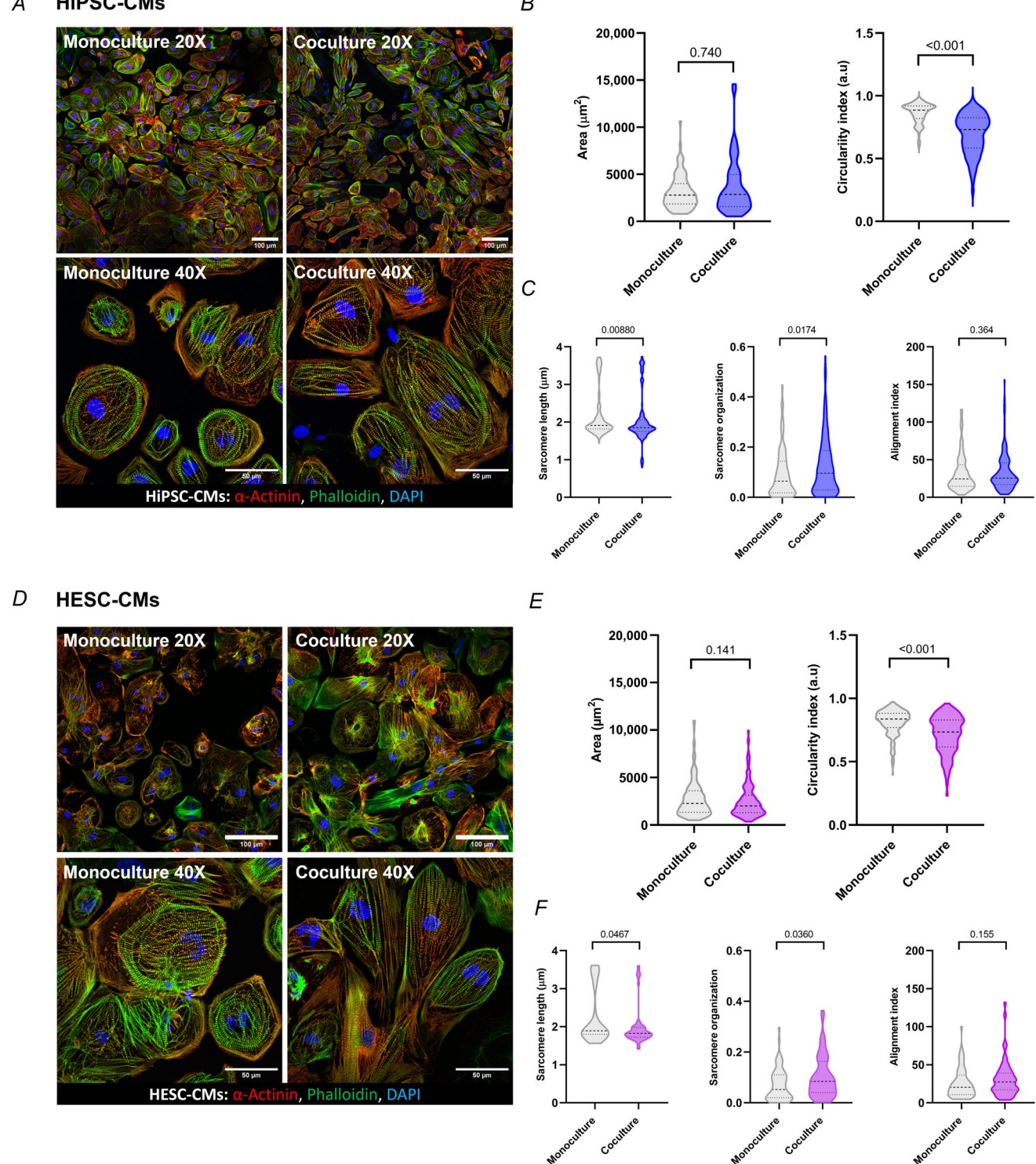

**Figure 4. SNs mature hPSC-CM cytoarchitecture**
Monocultures and cocultures were treated with 100 ng/ml NGF for 7–10 days and analysed for changes in cell structural parameters. *A*, representative immunofluorescence images of hiPSC-CMs for α-actinin (red), F-actin with phalloidin (green), and nuclei with DAPI (blue), in monoculture and coculture. 20× magnification (top) and 40× magnification images (bottom) were taken; with 40× for automated analysis of the structural characteristics of sarcomeres. Scale bar = 100 μm, 50 μm. *B*, no significant difference in the area of hiPSC-CMs in coculture ($P = 0.740$), but a significant reduction in the circularity index ($P < 0.001$) (monoculture *N/n* = 4/89, coculture

*N/n* = 4/82). *C*, significant reduction in sarcomere length (*P* = 0.00880), and significant increase in sarcomere organization (*P* = 0.0174), with no significant change in alignment index (*P* = 0.364) in cocultured (*N/n* = 8/147) compared to monocultured hiPSC-CMs (*N/n* = 8/154). *D*, representative images of hESC-CMs with the same staining as in *A* for comparison. *E*, in contrast to hiPSC-CMs, there was no significant change in area in coculture (*P* = 0.141). However, there was a significant reduction in circularity index (*P* < 0.001) in coculture (*N/n* = 4/255) compared to monoculture (*N/n* = 4/250) on hESC-CMs. *F*, significant reduction in sarcomere length (*P* = 0.0467), and significant increase in sarcomere organization (*P* = 0.0360), with no significant change in alignment index (*P* = 0.155) in cocultured (*N/n* = 4/55) compared to monocultured hESC-CMs (*N/n* = 4/63). Data presented as violin plots showing median with quartile range. *N/n* = biological replicate/cell number. [Colour figure can be viewed at wileyonlinelibrary.com]

Together, these data show SNs promoted improved $Ca^{2+}$ handling and contractile function within hPSC-CMs, with faster kinetics for departure velocity of both the contraction and $Ca^{2+}$ transients at baseline. hiPSC-CM AP changes towards a more adult-like phenotype further illustrated that SNs contribute to functional changes.

## Discussion

The ANS plays an essential role controlling cardiac contractility via the secretion of NA, which stimulates $\beta$AR at the surface of cardiomyocytes. Given that abnormalities in cardiac SNs contribute to various cardiac diseases, understanding the neuro-cardiac interaction is a rapidly emerging field. The complexity of this model has directed investigation into the study of sympathetic secretions on hPSC-CM maturation, focused development into the investigation of retrograde 'SN-to-CM' communication using bioengineering techniques, and the study of sympathoadrenergic diseases (Bernardin et al., 2022; Kowalski et al., 2022; Li et al., 2023). Here, we have successfully contributed to this field and provided a detailed analysis of the effect of SN stimulation on various aspects of hPSC-CM structure and function, with a particular focus on contractile activity, being a hallmark feature of CMs.

Critical to the success of these cocultures was the supplementation with NGF, which significantly upregulated neurite sprouting, promoting SN survival and growth in cocultures with hPSC-CMs. Interestingly, the 'pearl-necklace' morphology indicative of functional varicosities described by others (Prando et al., 2018), and the expansion of the neurite network can only be achieved by providing exogenous NGF during the 15 days tested in this work. This was not needed on the cocultures of SNs with NRVMs, in which the NGF release by the CMs is enough to sustain the network of SNs. In fact, the addition of exogenous NGF does not cause any significant improvement to this type of coculture (Dokshokova et al., 2022). Work from Bernardin et al. (Bernardin et al., 2022) has further shown that cocultures of hPSC-CMs with neurons can be obtained without the addition of exogenous NGF, although the experiments were performed after 42 days of coculture. This suggests a similar degree of neurite sprouting could have been achieved under our conditions without NGF, if we would have kept our cocultures for a longer period of time.

Next, we observed SNs could form synaptic-like connections with the hiPSC-CMs since SNAP-25-positive expression, a marker of neuroexocytosis, was found along both the soma and the neurites (Fig. 2). While coculture time increased the number of neurite branching points and overall surface area (Fig. 2), NCJs were observed with no discernible change in the size of the connection regardless of the coculture time (Fig. 3). This is in contrast to what Dokshokova et al. (2022) observed using NRVMs, where the NCJ increased from day 2 to day 15 of coculture. Our data indicate that SN neurite contacts present a higher value and variability even from the first day tested, probably caused by the exogenous NGF present in our conditions. However, most importantly, the significance of the SICM technique lies in the ability to scan only attached surfaces without touching the sample, meaning that physical anchorage of the neurite overlying the CM beneath needed to have taken place to be measured, as a floating neurite above the CM will disrupt image acquisition.

We then focused from the surface-level intercellular communication to the more downstream effects of SN stimulation on cytoplasmic machinery. Here, we found that there is a significant increase in the percentage of cytosolic cAMP of hiPSC-CMs in coculture compared to monoculture at increasing concentrations of nicotine, indicating that nicotine-induced NA release can stimulate the $\beta$AR response in cocultured hiPSC-CMs (Fig. 7). It was previously reported how $\beta$ARs are recruited to the CM surface in contact with the SNs, and how NA is released in a high local concentration in the vicinity of the synaptic cleft (Shcherbakova et al., 2007). While we could not determine if $\beta$ARs were recruited to the surface of hiPSC-CMs within these cocultures, our data indicate that nicotine-induced NA release elicited an increased $\beta$AR response within hiPSCs, indicative that the NCJs were functional. These results provide a solid ground for the investigation of contractile activity of CMs in response to SN stimulation. An interesting result is the higher response to IBMX in coculture (Fig. 7*D*), a global PDE blocker, which could indicate that SNs can modulate the activity of PDEs on the innervated CMs. Future work

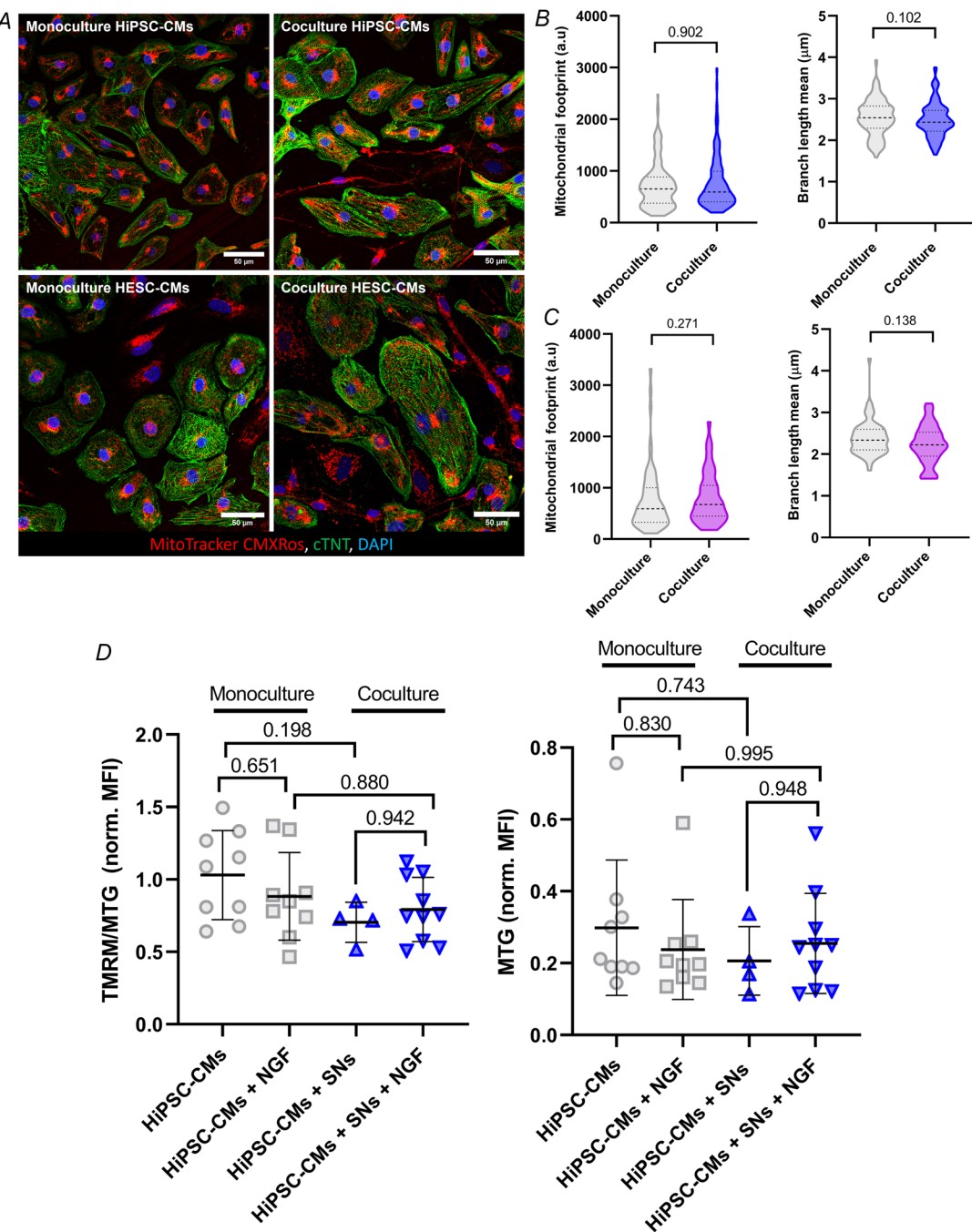

**Figure 5. Mitochondrial activity of hPSC-CMs is not affected in coculture**
Monocultures and cocultures were treated with 100 ng/ml NGF for 7–10 days and analysed for changes in mitochondrial activity. *A*, representative images of MitoTracker CMXRos (red) used for live immunostaining of mitochondrial networks in hiPSC-CMs (top) and hESC-CMs (bottom) prior to fixation and staining with cTNT (green), and nuclei with DAPI (blue). Scale bar = 50 μm. *B*, no significant change in mitochondrial footprint (*P* = 0.902) or mean branch length (*P* = 0.102) of hiPSC-CMs in monoculture *versus* coculture (*N/n* = 6/190 in monoculture, *N/n* = 5/126 in coculture). *C*, no significant change in mitochondrial footprint (*P* = 0.271) or mean branch length (*P* = 0.138) of hESC-CMs in monoculture *versus* coculture (*N/n* = 2/64 in monoculture, *N/n* = 2/47 in coculture). Data presented as violin plots showing median with quartile range, Mann–Whitney test. *D*, flow cytometry analysis in hiPSC-CMs showed no significant difference in tetramethylrhodamine, methyl ester (TMRM) or MitoTracker Green (MTG), used as a measure of mitochondrial membrane potential and abundance, respectively; normalized to mean fluorescence intensity (MFI). Data presented as mean with SD, one-way ANOVA, *N* ≥ 4. *N/n* = biological replicate/cell number. [Colour figure can be viewed at wileyonlinelibrary.com]

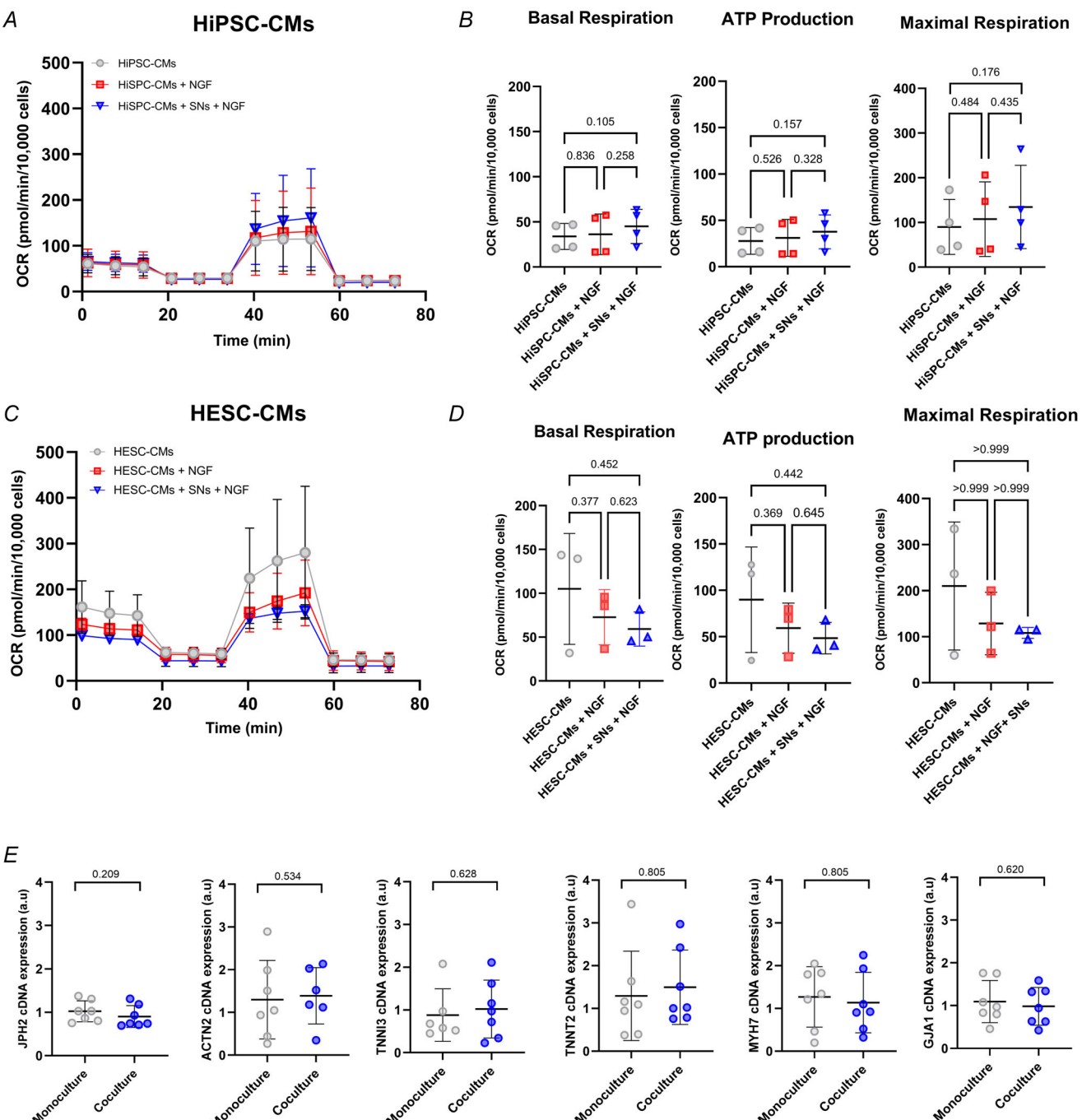

**Figure 6. No significant change in respiration or structural gene expression of CMs in coculture**
The Seahorse XF test was performed to measure mitochondrial respiration and parameters of oxygen consumption rate (OCR). The effect of NGF on hPSC-CM respiration was investigated independently. *A* and *B*, there was no significant difference in OCR across all three conditions with respect to basal respiration, ATP production or maximal respiration in hiPSC-CMs. Data presented as mean with SD, one-way ANOVA (except maximal respiration; Kruskal–Wallis test), $N = 4$. *C* and *D*, there was no significant difference in OCR across all three conditions with respect to basal respiration, ATP production or maximal respiration in hESC-CMs. Data presented as mean with SD, one-way ANOVA, $N = 3$. *E*, RT-qPCR confirmed a lack of significant change in the expression of key structural genes involved in hiPSC-CM contraction. Data presented as mean with SD, Mann–Whitney test, $N \geq 6$. $N$ = biological replicate. [Colour figure can be viewed at wileyonlinelibrary.com]

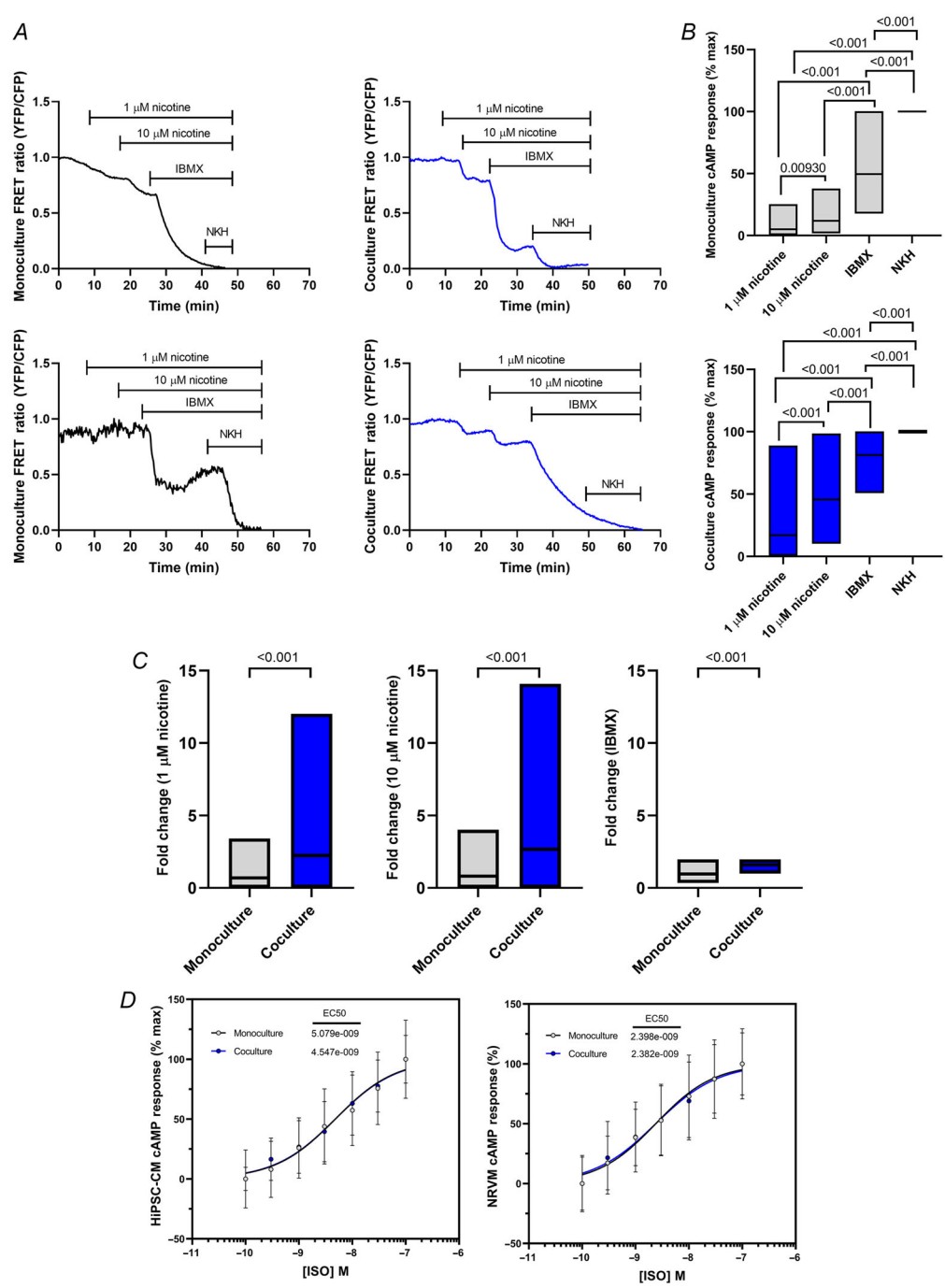

**Figure 7. cAMP production in cocultured hiPSC-CMs is larger than monoculture in response to nicotine**
*A*, representative traces of FRET response from monoculture and coculture, to 1 and 10 µM nicotine, 100 µM of the PDE inhibitor IBMX, and 10 µM of the adenylyl cyclase activator NKH, to normalize to the maximum cAMP response. Note that upon binding to cAMP, the sensor exhibits a decrease in FRET. YFP, yellow fluorescent protein; CFP, cyan fluorescent protein. *B*, summary graphs showing cAMP production in monoculture (*N/n* = 5/58; *P* < 0.001, *P* = 0.00930) and coculture (*N/n* = 5/69; *P* < 0.001). Data presented as mean with SD, with Friedman test. *C*, summary graphs comparing monoculture and coculture at each drug application (1 µM nicotine, 10 µM nicotine, IBMX), normalized to the response in monoculture. Data presented as mean with SD, *P* < 0.001, Mann–Whitney test (except IBMX; unpaired *t* test). *D*, concentration–response curves depict the increase in response to iso-proterenol (ISO), confirming the presence of functional βARs in hiPSC-CMs and NRVMs. There was no significant difference in hiPSC-CMs (monoculture *N/n* = 6/63; coculture *N/n* = 7/69) or in NRVMs (monoculture *N/n* = 3/32; coculture *N/n* = 2/39). Data presented as mean with SD. *N/n* = biological replicate/cell number. [Colour figure can be viewed at wileyonlinelibrary.com]

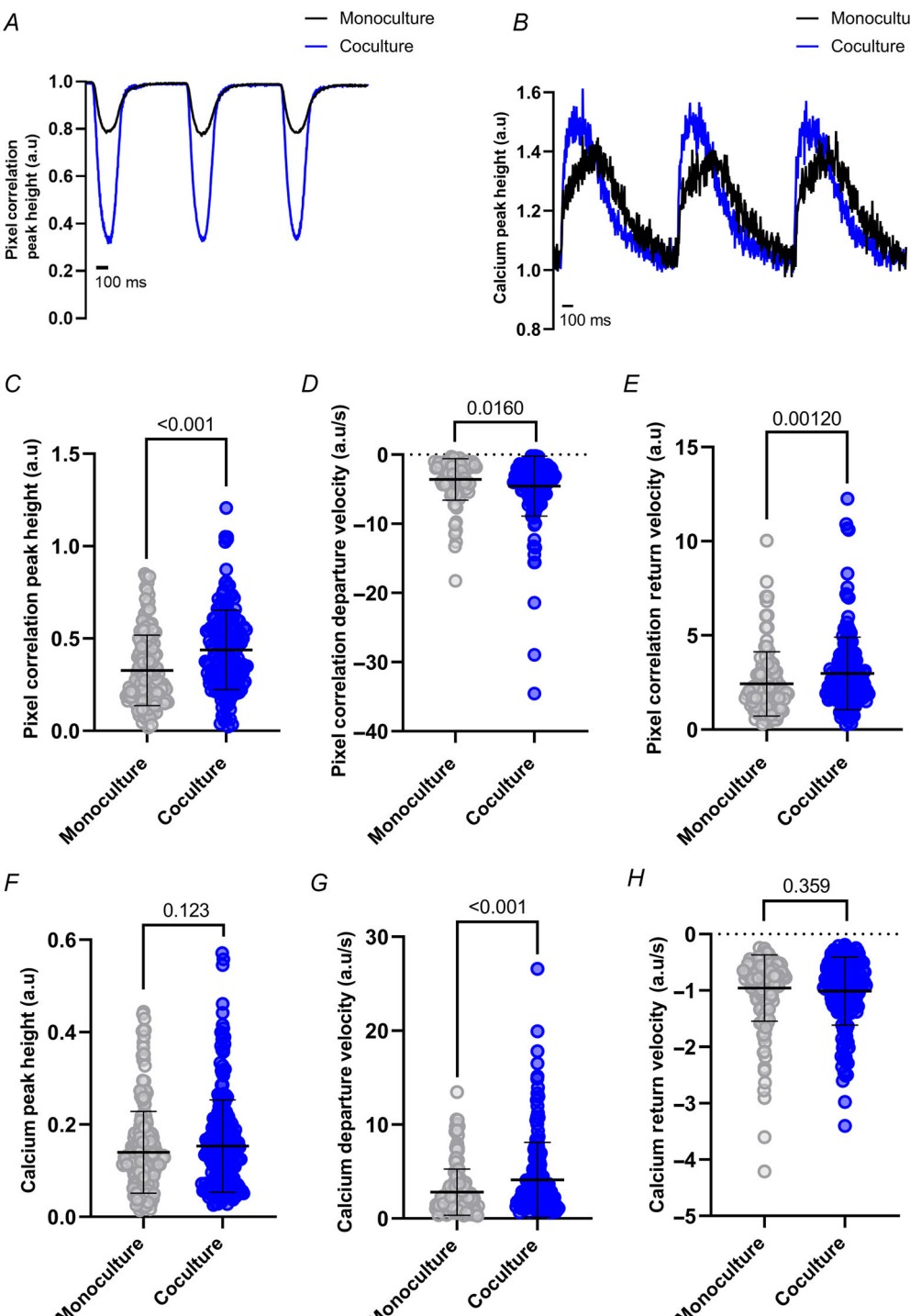

**Figure 8. Baseline contraction and Ca²⁺ transient analysis shows larger activity in cocultured hiPSC-CMs**
The CytoCypher was used to simultaneously measure $Ca^{2+}$ and contractile activity of hiPSC-CMs. Representative traces at baseline of *A*, pixel correlation and *B*, $Ca^{2+}$ transients. A summary of results is shown with monocultures represented in grey, and cocultures in blue. *C*, pixel correlation peak height ($P < 0.001$) was significantly larger in coculture ($N/n$ = 15/174) than monoculture ($N/n$ = 12/160), and *D*, departure velocity ($P = 0.0160$) and *E*, return velocity ($P = 0.00120$) were significantly faster in coculture ($N/n$ = 15/170) than monoculture ($N/n$ = 11/107). *F*, $Ca^{2+}$ transient peak height ($P = 0.123$) was larger in coculture ($N/n$ = 15/250) than monoculture ($N/n$ = 14/242), but not statistically significant. *G*, departure velocity ($P < 0.001$) was significantly faster in coculture ($N/n$ = 10/174) than monoculture ($N/n$ = 10/173), whilst *H*, return velocity ($P = 0.359$) was not. Data presented as mean with SD, Mann–Whitney test. $N/n$ = biological replicate/cell number. [Colour figure can be viewed at wileyonlinelibrary.com]

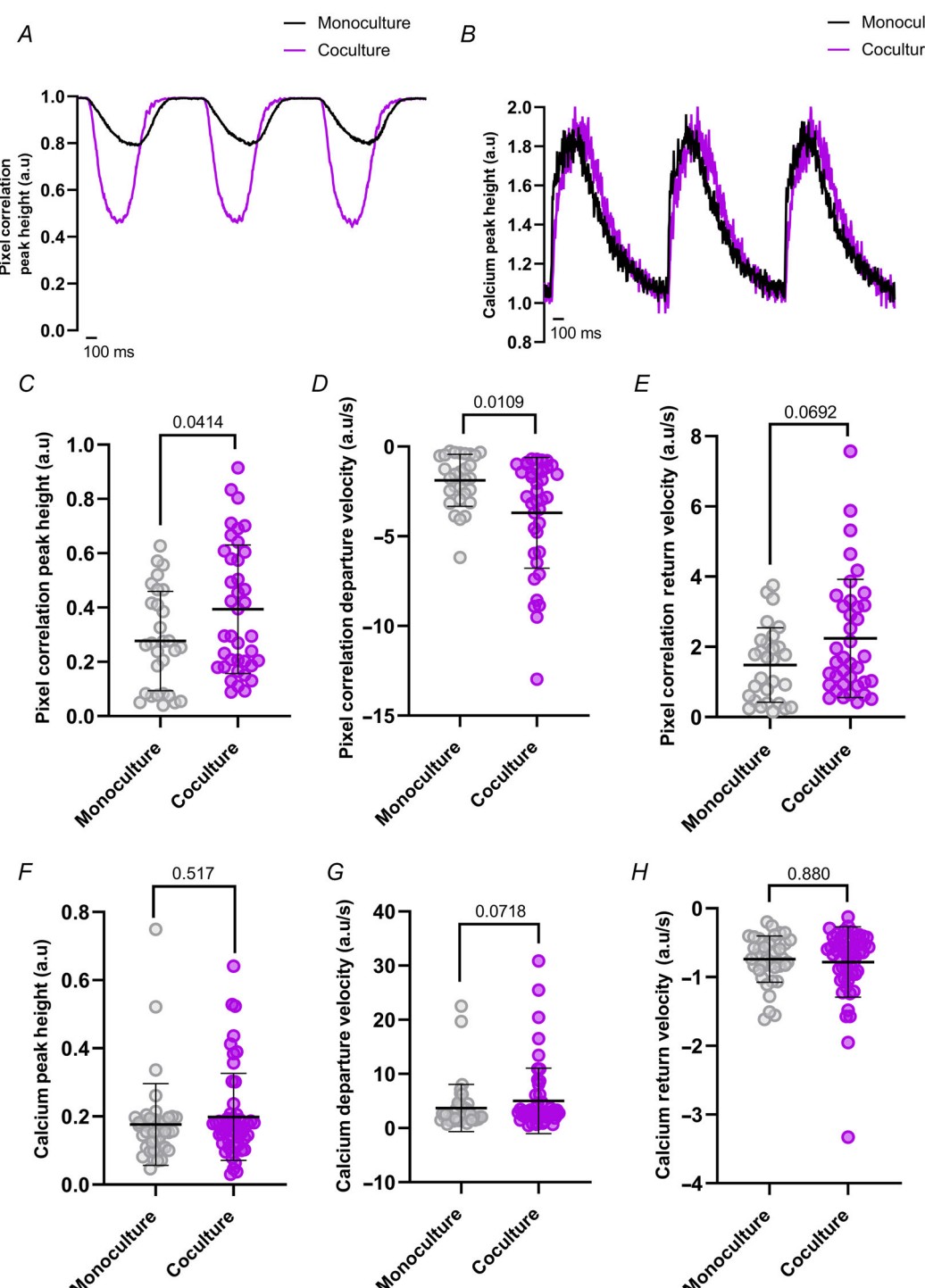

**Figure 9. Baseline contraction and Ca²⁺ transient analysis shows larger activity in cocultured hESC-CMs**
The CytoCypher was used to simultaneously measure Ca²⁺ and contractile activity of hESC-CMs. Representative traces at baseline of *A*, pixel correlation and *B*, Ca²⁺ transients. A summary of results is shown with monocultures represented in grey, and cocultures in purple. *C*, pixel correlation peak height (*P* = 0.0414) was significantly larger in coculture (*N/n* = 4/39) than monoculture (*N/n* = 4/29), and *D*, departure velocity (*P* = 0.0109) was significantly faster in coculture, whilst *E*, return velocity (*P* = 0.0692) was not significantly changed (*N/n* = 4/38) compared to monoculture (*N/n* = 4/29). *F*, Ca²⁺ transient peak height (*P* = 0.517) was not significantly affected between monoculture (*N/n* = 4/42) and coculture (*N/n* = 4/53). Similarly, *G*, departure velocity (*P* = 0.0718) and *H*, return velocity (*P* = 0.880) were not significantly affected in monoculture (*N/n* = 4/41) and coculture (*N/n* = 4/54). Data presented as mean with SD, Mann–Whitney test. *N/n* = biological replicate/cell number. [Colour figure can be viewed at wileyonlinelibrary.com]

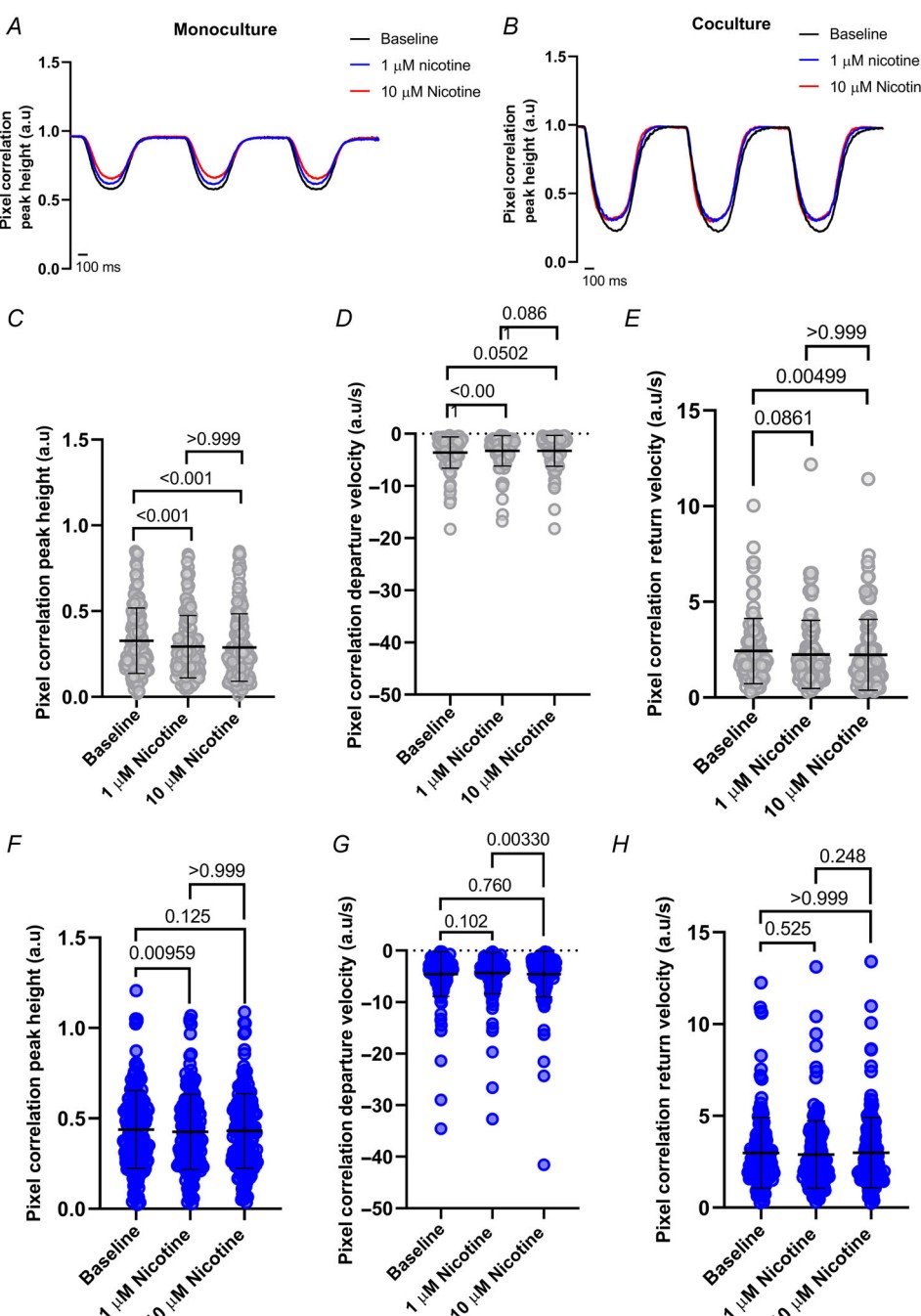

**Figure 10. The pixel correlation response to nicotine is rescued in cocultured hiPSC-CMs**
Representative traces show stimulation of cells after 1 and 10 μM nicotine following baseline measurements in *A*, monoculture and *B*, coculture. hiPSC-CMs in monoculture are depicted in grey, and coculture in blue. *C*, pixel correlation peak height in monoculture (*N/n* = 12/160) was significantly reduced from baseline after 1 and 10 μM nicotine (*P* < 0.001), with no significant difference between both nicotine applications (*P* > 0.999). *D*, departure velocity (*N/n* = 11/107) was significantly slower after 1 μM nicotine (*P* < 0.001), with no change after 10 μM nicotine (*P* = 0.0502, *P* = 0.0861). *E*, return velocity (*N/n* = 11/107) was significantly slower than baseline after 10 μM nicotine (*P* = 0.00499), but not after 1 μM nicotine (*P* = 0.0861), or between both nicotine applications (*P* > 0.999). *F*, in coculture, the pixel correlation peak height (*N/n* = 15/174) was significantly reduced after 1 μM nicotine (*P* = 0.00959), which recovered after nicotine (*P* = 0.125, *P* > 0.999). *G*, departure velocity (*N/n* = 15/170) in cocultured hiPSC-CMs was not significantly affected (*P* = 0.102, *P* = 0.760), except after 10 μM nicotine (*P* = 0.00330). *H*, pixel correlation return velocity (*N/n* = 15/170) was not significantly affected after nicotine (*P* = 0.525, *P* > 0.999, *P* = 0.248). Data presented as mean with SD, Friedman test. *N/n* = biological replicate/cell number. [Colour figure can be viewed at wileyonlinelibrary.com]

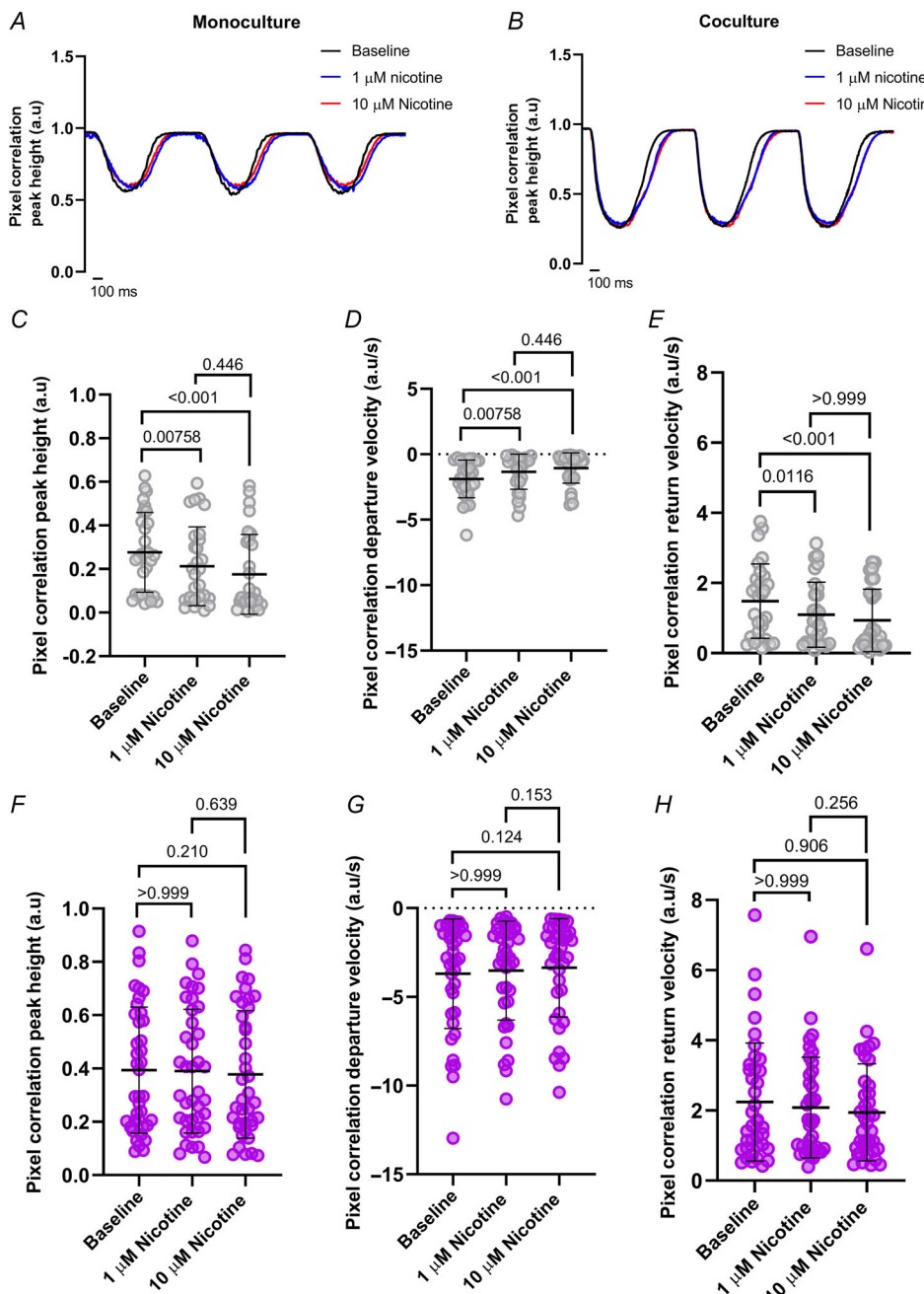

**Figure 11. The pixel correlation response to nicotine is rescued in cocultured hESC-CMs**
Representative traces show stimulation of cells after 1 and 10 μM nicotine following baseline measurements in *A*, monoculture and *B*, coculture. hiPSC-CMs in monoculture are depicted in grey, and coculture in purple. *C*, pixel correlation peak height in monoculture (*N/n* = 4/29) was significantly reduced from baseline after 1 μM nicotine ($P = 0.00758$) and 10 μM nicotine ($P < 0.001$), with no significant difference between both nicotine applications ($P = 0.446$). *D*, departure velocity (*N/n* = 4/29) was significantly slower after 1 μM nicotine ($P = 0.00758$) and 10 μM nicotine ($P < 0.001$), with no significant difference between both nicotine applications ($P = 0.446$). *E*, return velocity (*N/n* = 4/29) was significantly slower than baseline after 1 μM nicotine ($P = 0.0116$) and 10 μM nicotine ($P < 0.001$), but not significantly affected between both nicotine applications ($P > 0.999$). *F*, in coculture, the pixel correlation peak height (*N/n* = 4/39) in cocultured hiPSC-CMs was not significantly affected from baseline after nicotine ($P > 0.999$, $P = 0.210$, $P = 0.639$). *G*, departure velocity (*N/n* = 4/38) in cocultured hiPSC-CMs was not significantly affected from baseline after nicotine ($P > 0.999$, $P = 0.124$, $P = 0.153$). *H*, pixel correlation return velocity (*N/n* = 4/38) was not significantly affected after nicotine ($P > 0.999$, $P = 0.906$, $P = 0.256$). Data presented as mean with SD, Friedman test. *N/n* = biological replicate/cell number. [Colour figure can be viewed at wileyonlinelibrary.com]

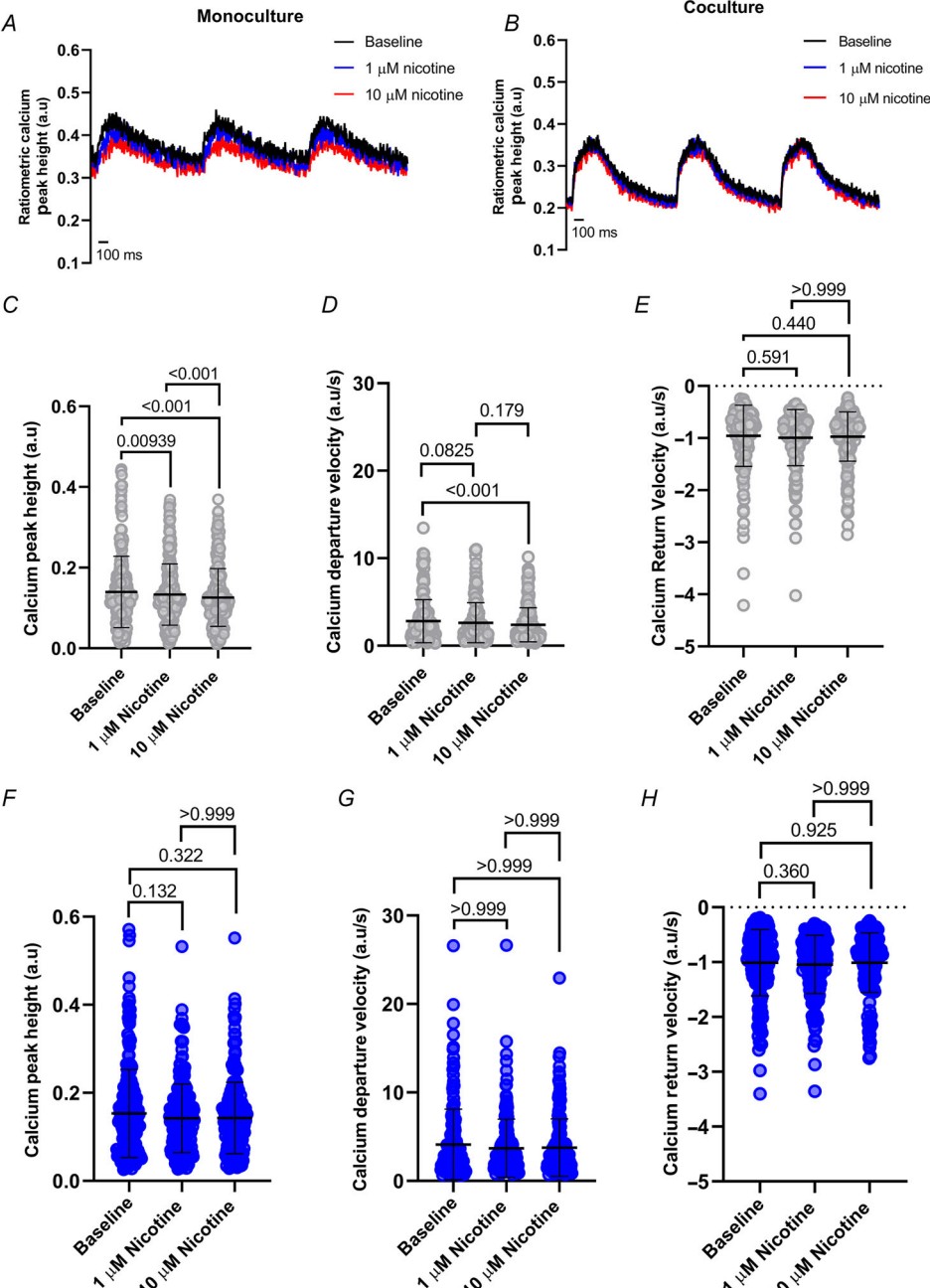

**Figure 12. The Ca²⁺ transient response to nicotine is rescued in cocultured hiPSC-CMs**

Representative traces show stimulation of cells after 1 and 10 μM nicotine following baseline measurements in *A*, monoculture and *B*, coculture. hiPSC-CMs in monoculture are depicted in grey, and coculture in blue. *C*, Ca²⁺ transient peak height in monoculture (*N/n* = 14/242) was significantly reduced from baseline after 1 μM nicotine ($P = 0.00939$) and 10 μM nicotine ($P < 0.001$), which significantly dropped between both nicotine applications ($P < 0.001$). *D*, departure velocity (*N/n* = 10/173) was not significantly affected after 1 μM nicotine ($P = 0.0825$), but was significantly slower after 10 μM nicotine ($P < 0.001$), with no significant change between both applications ($P = 0.179$). *E*, return velocity (*N/n* = 10/173) was not significantly affected after 1 μM nicotine ($P = 0.591$) or 10 μM nicotine ($P = 0.440$), nor between both applications ($P > 0.999$). *F*, in coculture, the Ca²⁺ transient peak height (*N/n* = 15/250) was not significantly affected after 1 μM nicotine ($P = 0.132$) or 10 μM nicotine ($P = 0.322$), nor between both applications ($P > 0.999$). *G*, departure velocity (*N/n* = 10/174) in cocultured hiPSC-CMs was not significantly affected from baseline ($P = 0.132$, $P = 0.322$), nor between both applications ($P > 0.999$). Similarly, *H*, return velocity (*N/n* = 10/174) was not significantly affected after baseline ($P = 0.360$, $P = 0.925$) nor between both applications ($P > 0.999$). Data presented as mean with SD, Friedman test. *N/n* = biological replicate/cell number. [Colour figure can be viewed at wileyonlinelibrary.com]

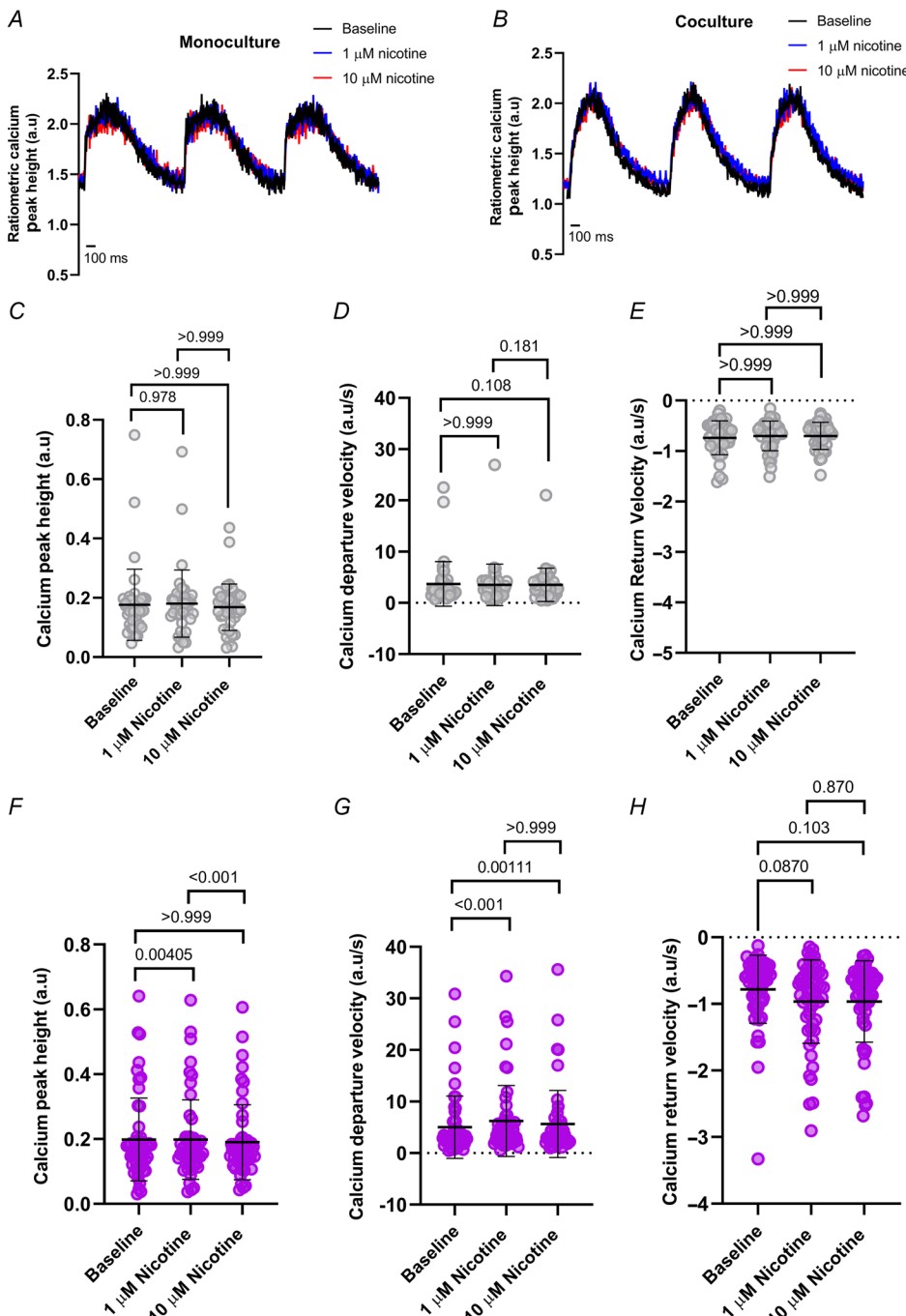

**Figure 13. The Ca²⁺ transient response to nicotine is rescued in cocultured hESC-CMs**

Representative traces show stimulation of cells after 1 and 10 μM nicotine following baseline measurements in *A*, monoculture and *B*, coculture. hESC-CMs in monoculture are depicted in grey, and coculture in purple. *C*, Ca²⁺ transient peak height in monoculture (*N/n* = 4/42) was not significantly affected after nicotine ($P = 0.978$, $P > 0.999$). *D*, departure velocity (*N/n* = 4/41) was not significantly affected after nicotine ($P > 0.999$, $P = 0.108$, $P = 0.181$). *E*, return velocity (*N/n* = 4/41) was not significantly affected after nicotine ($P > 0.999$). *F*, in coculture, the Ca²⁺ transient peak height (*N/n* = 4/53) was significantly increased after 1 μM nicotine ($P = 0.00405$), but was significantly reduced between both nicotine applications ($P < 0.001$). There was no significant change from baseline after 10 μM nicotine ($P > 0.999$). *G*, departure velocity (*N/n* = 4/54) in cocultured hESC-CMs was significantly faster than baseline after 1 μM nicotine ($P < 0.001$) and 10 μM nicotine ($P < 0.00111$), but unchanged between both applications ($P > 0.999$). *H*, return velocity (*N/n* = 4/54) was not significantly affected after nicotine ($P = 0.0870$, $P = 0.103$, $P = 0.870$). Data presented as mean with SD, Friedman test. *N/n* = biological replicate/cell number. [Colour figure can be viewed at wileyonlinelibrary.com]

on this hypothesis will provide a better understanding of neuro-cardiac communication.

Prior to the investigation of contraction and $Ca^{2+}$ handling, imaging was used to determine any changes in the structure and cell cytoarchitecture that could affect these functions. The most successful results on structural changes on pluripotent stem cells have been observed through time-dependent maturation in culture (Kamakura et al., 2013; Lundy et al., 2013; Snir et al., 2003). Interestingly, Kowalski et al. (2022) found a significant increase in sarcomere organization, and a significant decrease in sarcomere length by coculturing stem cells and mouse SNs. We can confirm the same with our models of CMs and rat SNs (Fig. 4), as well as an increase in cellular elongation, in hPSC-CMs. These results strengthen the

hypothesis that SNs induce CM structural maturity, although the exact mechanism by which this has been achieved is not yet understood. We did not observe any mRNA expression changes for the genes tested in this study (Fig. 6). Although this does not rule out a possible increase in protein levels, our findings on sarcomere organization and functional changes suggest that the improved cytoarchitecture observed in the 7–10 day cocultures is a consequence of an enhanced cellular function induced by the SNs. Another possibility may be that the physical anchoring of SNs onto the CMs triggers mechanical cues. Important scaffolding proteins have been shown to be upregulated in the context of mouse ventricular myocyte cocultures, such as AKAP79/150 and synapsin-1, which could help explain this phenomenon

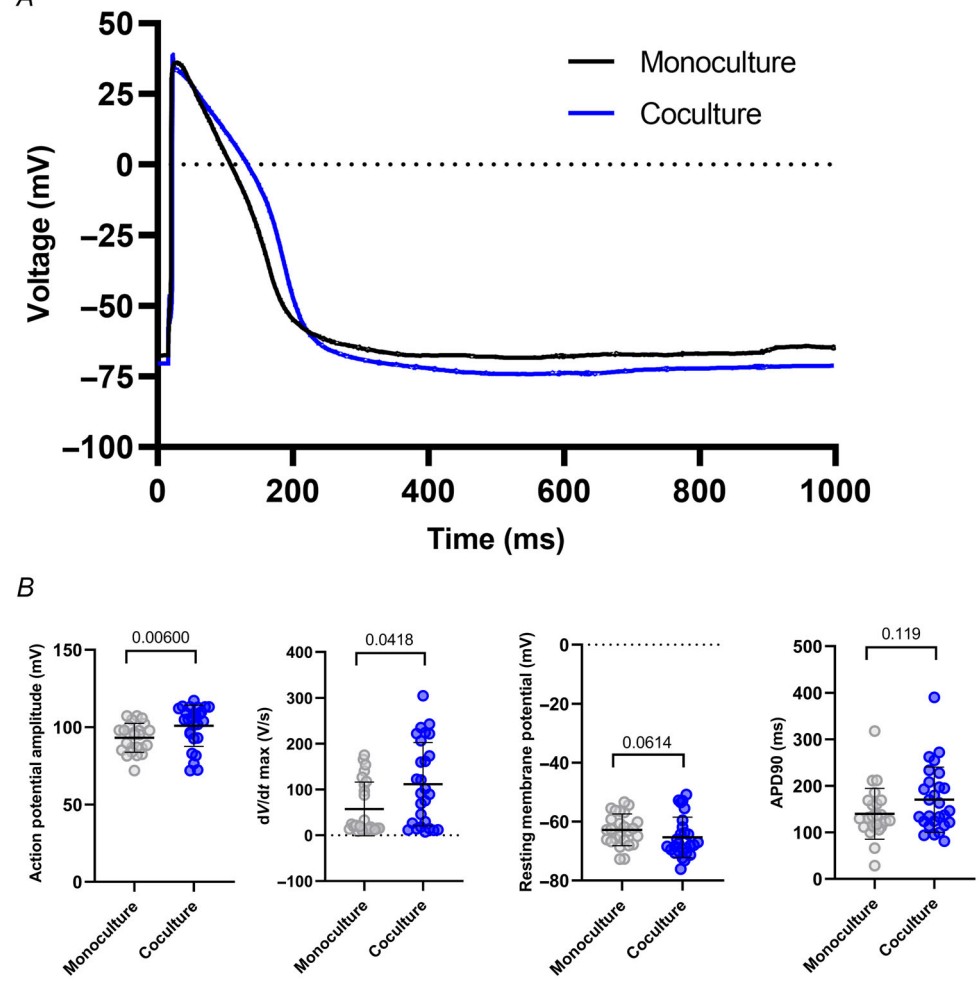

**Figure 14. Electrophysiological characteristics are significantly matured in cocultured hiPSC-CMs**
*A*, representative trace of patch-clamp recordings of individual hiPSC-CMs. Summary graphs show hiPSC-CMs in monoculture (*N*/*n* = 3/25) in grey, and coculture (*N*/*n* = 3/27) in blue. *B*, significant increase in action potential amplitude (*P* = 0.00600) in coculture. Significant increase in d*V*/d*t* max (maximum change in voltage rate) in coculture (*P* = 0.0418). No significant difference in resting membrane potential (*P* = 0.0614). No significant difference in action potential duration 90 (APD90) between monoculture and coculture. Data presented as mean with SD, Mann–Whitney test. *N*/*n* = biological replicate/cell number. [Colour figure can be viewed at wileyonlinelibrary.com]

(Shcherbakova et al., 2007). An increase in cAMP levels has also been linked to enhanced maturation of hPSC-CMs, including improved sarcomere organization and greater amplitude and upstroke velocity of the AP (Giacomelli et al., 2020). Although FRET cannot directly report baseline cAMP levels in our case, it is plausible that SNs may alter baseline cAMP levels and further promote maturation.

Our most prominent finding is that SNs modulate functional properties, i.e. ion channel activity, contractile function and $Ca^{2+}$ handling of hPSC-CMs. It has already been shown that neurons can modulate sarcoplasmic reticulum calcium handling and contraction in hPSC-CMs (Bernardin et al., 2022). Specifically, the presence of PC12 neurons was found to increase calcium release velocity at baseline. Similarly, we found an increase in the contraction and $Ca^{2+}$ transient amplitudes of hPSC-CMs at baseline (Figs 8 and 10). These findings align with the increase in sarcomere organization, which should enable enhanced force–strength relationships in the CMs. The increase in amplitude at baseline and in coculture is consistent with Kowalski et al. (2022), who surprisingly found that the 90% return time was unexpectedly longer than in monoculture. The authors suggest that SNs could have a bi-directional effect on the development of CM $Ca^{2+}$ transients, with a more mature phenotype with respect the transient peak, but simultaneously more immature for the time of the transient. However, under our conditions, SNs improve the CM $Ca^{2+}$ handling properties in an equal fashion. This discrepancy could reflect the fact that Kowalski et al. (2022) evaluated contraction within spontaneous beating cultures while we paced the cultures at a frequency of 1 Hz. Cocultures show a higher transient peak and faster kinetics; these results also correlate with our findings in the contraction properties of these cells, measured as pixel correlation, showing a higher peak amplitude and faster kinetics when CMs are cocultured with SNs. These data are indicative of a more mature phenotype in the presence of SNs. The advantage of our pacing approach is that we can directly compare $Ca^{2+}$ peak transient at the same frequency of beating (1 Hz), avoiding any confounding effect of the beating frequency itself on the transient, as an increase in beating frequency decreases the $Ca^{2+}$ transient peak, and vice versa. We also obtained similar results to Narkar et al. (2022), who demonstrated a significant increase in contraction amplitude and faster contraction kinetics in coculture *versus* monoculture under paced conditions.

Finally, we have also shown a significant increase in AP amplitude and its depolarization rate in cocultured hiPSC-CMs (Fig. 14). Both changes are indicative of a more adult-like AP phenotype. Similar changes in AP amplitude and depolarization rate have been previously reported in cocultures of hiPSC-CMs with cardiac fibroblasts (Giacomelli et al., 2020). Our study demonstrates that SNs can positively influence the electrophysiological properties of hiPSC-CMs. This could indicate that innervation of hPSC-CMs could trigger changes in ion channel expression, or their translocation to the surface membrane. An increase of the amplitude and the depolarization rate may be attributable to an increase of the $Na^+$ channel currents in the cocultured hiPSC-CMs.

## Limitations of the study

A limitation of our study is that the CMs were losing contraction capability over the course of the experiments. Most probably, the conditions of the experiments themselves, pre-loading the cells with Fura-AM, the use of HBSS at 37°C and the constant 1 Hz pacing, could collectively trigger the degradation of the contraction machinery. As observed in the recordings taken in monoculture, the peak height of both pixel correlation and $Ca^{2+}$ transients, as well as their kinetics, were decreased over time (Figs 10–13). However, despite this, nicotine can still trigger an inotropic and chronotropic effect on the cocultured CMs, when compared to the negative monoculture decay at the same time points. Nevertheless, more optimal experimental conditions for the CMs in this context would have benefitted the results considerably. Another limitation is that we have not monitored neurotransmitter release within the neuro-cardiac junction space. It has been described that the synaptic cleft between neurons and cardiomyocytes is a relatively narrow intercellular space (80–100 nm wide) where noradrenaline (NA) is preferentially released due to the polarization of the neuronal active zone (Zaglia & Mongillo, 2017). This structural arrangement allows for a high local concentration of NA upon the release of relatively few molecules, efficiently activating cardiac $\beta$ARs. Consequently, even if low NA levels are detected in the culture medium, as demonstrated by Bernardin et al. (2022) for acetylcholine in cocultures, this would not accurately reflect the localized effects of sympathetic neurons on cardiomyocytes. To fully validate our experiments, a technical breakthrough is required to enable real-time measurement of NA release within the synaptic cleft.

## Conclusions

We have found that modelling the NCJ through direct cell–cell contact *in vitro* can induce substantial structural and functional effects on CMs. Cell elongation in coculture indicates maturation of the cell cytoarchitecture, which probably contributes to force–frequency relationships vital for a key hallmark of

the CM phenotype – contraction. Next, we delved deeper into the cytosolic signalling cascades largely using nicotine to induce an NA-dependent $\beta$AR response. Importantly, we investigated for the first time the effects of sympathetic stimulation on two major pathways of adult CMs also present in hiPSC-CMs, including the cAMP/protein kinase A pathway and CICR (calcium-induced calcium release) necessary for contraction/Ca$^{2+}$ handling. We show that SNs strongly contribute to cAMP production and enlarge contraction/Ca$^{2+}$ transient amplitudes at baseline. Interestingly, this maturity axis does not elicit metabolic maturation, at least in the time frame of these experiments. Together, these findings offer new insights into the role SNs play in cardiomyocyte development and this coculture model represents an exciting new possibility into the study of sympathoadrenergic disease, and may also contribute to developments in cardiac tissue engineering through the successful establishment of cell–cell interactions in coculture.

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

## Additional information

### Data availability statement

All the data are available from the corresponding authors upon reasonable request.

## Competing interests

No competing interests declared.

## Author contributions

N.M. prepared all the monocultures and cocultures for the study, performed the majority of the experiments, analysed the data and interpreted the results. L.F. performed electrophysiological experiments and critically discussed the data. P.C. grew cardiomyocytes for experiments, performed contraction experiments and cellular architecture experiments, and analysed the data. V.L. performed electrophysiological experiments and analysed the data. L.T. grew cardiomyocytes for experiments, performed metabolic experiments and analysed the data. H.G. and S.S. analysed the data. M-V.C. supported the stem cell differentiation and interpreted the results. J.L.S-A. performed SICM experiments, analysed the data and interpreted the results. N.M. and J.L.S-A wrote the manuscript with help from L.F. A.S.B., J.G. and J.L.S-A. designed and supervised the study. All authors revised the manuscript.

## Funding

This work was supported by British Heart Foundation (grant RG/F/22/110081 to J.G./J.S.A. and FS/IBSRF/23/25188 to J.S.A.). Work in the lab of A.S.B. was supported by a Wellcome Trust career re-entry fellowship (210987/Z/18/Z), an idea to innovation grant from the Francis Crick Institute (supported by MR/X50287X/1), and a LifeArc grant (LifeArc/Crick Translation Fund).

## Acknowledgements

We are grateful to the Facility for Imaging by Light Microscopy (FILM) and the Centre of Excellence Cellular Mechanosensing and Functional Microscopy at Imperial College London. We thank the following scientific platforms/units at the Francis Crick Institute: the human embryo and stem cell unit, and the flow cytometry facility for their expertise, support and use of facilities. We especially thank Lyn Healy from the human embryo and stem cell unit for advice and support with stem cells. We are grateful to Imperial College London for the Presidents PhD Scholarship of Neda Mohammadi.

## Keywords

cardiomyocyte (CM), $Ca^{2+}$, hPSC, pluripotency, sympathetic neuron (SN)

## Supporting information

Additional supporting information can be found online in the Supporting Information section at the end of the HTML view of the article. Supporting information files available:

**Peer Review History**

