## [Peer Review History · The Journal of Physiology]

Sympathetic Neurons can modify the intrinsic structural and functional properties of hPSC-CMs

Neda Mohammadi, Laura Fedele, Poornaa Chakravarthy, Vladislav Leonov, Lorenza Tsansizi, Hui Gu, Sama Seyedmousavi, Marie-Victoire Cosson, Andreia Sofia Bernardo, Julia Gorelik, and Jose L. Sanchez-Alonso
DOI: 10.1113/JP287569

Corresponding author(s): Jose Sanchez-Alonso (j.sanchez-alonso-mardones@imperial.ac.uk)

Review Timeline:

Submission Date:	29-Aug-2024
Editorial Decision:	23-Sep-2024
Revision Received:	29-Dec-2024
Editorial Decision:	22-Jan-2025
Revision Received:	29-Jan-2025
Accepted:	31-Jan-2025

Senior Editor: Harold Schultz

Reviewing Editor: T Alexander Quinn

Transaction Report:

Dear Dr Sanchez-Alonso,

Re: JP-RP-2024-287569 "Sympathetic Neurons can modify the intrinsic structural and functional properties of hiPSC-CMs" by Neda Mohammadi, Laura Fedele, Vladislav Leonov, Lorenza Tsansizi, Hui Gu, Sama SeyedMousavi, Poornaa Chakravarthy, Andreia Sofia Bernardo, Julia Gorelik, and Jose L. Sanchez-Alonso

Thank you for submitting your manuscript to The Journal of Physiology. It has been assessed by a Reviewing Editor and by 2 expert referees and we are pleased to tell you that it is potentially acceptable for publication following satisfactory major revision.

LANGUAGE EDITING AND SUPPORT FOR PUBLICATION: If you would like help with English language editing, or other article preparation support, Wiley Editing Services offers expert help, including English Language Editing, as well as translation, manuscript formatting, and figure formatting at www.wileyauthors.com/eoo/preparation. You can also find resources for Preparing Your Article for general guidance about writing and preparing your manuscript at www.wileyauthors.com/eoo/prepresources.

REVISION CHECKLIST:

We look forward to receiving your revised submission.

Yours sincerely,

Harold Schultz
Senior Editor
The Journal of Physiology

REQUIRED ITEMS

- Author photo and profile. First or joint first authors are asked to provide a short biography (no more than 100 words for one author or 150 words in total for joint first authors) and a portrait photograph. These should be uploaded and clearly labelled together in a Word document with the revised version of the manuscript. See Information for Authors for further details.

- You must start the Methods section with a paragraph headed Ethical approval (https://jp.msubmit.net/cgi-bin/main.plex?form_type=display_requirements#methods).

Research must comply with The Journal's policies regarding animal experiments (<https://physoc.onlinelibrary.wiley.com/hub/animal-experiments>) and adherence to these policies must be stated in the manuscript.

Authors should confirm in their Methods section that their experiments were carried out according to the guidelines laid down by their institution's animal welfare committee, including an ethics approval reference number. The Methods section must contain a statement about access to food, water and housing, details of the anaesthetic regime: anaesthetic used, dose and route of administration, and method of killing the experimental animals.

- The reference list must be in alphabetical order, rather than numbered, to comply with our Journal format.

- Your manuscript must include a complete Additional Information section, including competing interests; funding; author contributions and acknowledgements.

- Please upload separate high-quality figure files via the submission form.

- Please ensure that any tables are editable and in Word format, and wherever possible, embedded in the article file itself.

- Please ensure that the Article File you upload is a Word file.

- Your paper contains Supporting Information of a type that we no longer publish, including supplementary tables and figures. Any information essential to an understanding of the paper must be included as part of the main manuscript and

figures. The only Supporting Information that we publish are video and audio, 3D structures, program codes and large data files. Your revised paper will be returned to you if it does not adhere to our Supporting Information Guidelines.

- Please include an Abstract Figure file, as well as the Figure Legend text within the main article file. The Abstract Figure is a piece of artwork designed to give readers an immediate understanding of the research and should summarise the main conclusions. If possible, the image should be easily 'readable' from left to right or top to bottom. It should show the physiological relevance of the manuscript so readers can assess the importance and content of its findings. Abstract Figures should not merely recapitulate other figures in the manuscript. Please try to keep the diagram as simple as possible and without superfluous information that may distract from the main conclusion(s). Abstract Figures must be provided by authors no later than the revised manuscript stage and should be uploaded as a separate file during online submission labelled as File Type 'Abstract Figure'. Please also ensure that you include the figure legend in the main article file. All Abstract Figures should be created using BioRender. Authors should use The Journal's premium BioRender account to export high-resolution images. Details on how to use and access the premium account are included as part of this email.

EDITOR COMMENTS

Reviewing Editor:

Your paper has been reviewed by two experts in the field. They felt the studies comprised a thorough experimental approach, which has led to the compelling and important finding that neuronal stimulation leads to increased cytosolic cAMP in hiSPC-CM, which enhances their function and maturity. However, they also felt that some of the work was premature in nature, such that conclusions were not well-supported by the experimental results, thus requiring additional experiments and analyses along with a major revision of the paper as outlined in the Comments for the Author. If you are willing to conduct this additional work and make the required revisions, please submit a revised manuscript, along with a point-by-point response to the reviewers' concerns.

Please also see 'Required Items' above.

Senior Editor:

Thank you for submission of your research article to the Journal of Physiology for consideration. The article has been reviewed by experts in the field and found to be potentially acceptable for publication pending a major revision to address all of the concerns raised. It was deemed that additional experiments are needed to address some experiments found to be premature for conclusions raised as outlined in the reviews. Please address all comments from the external referees and reviewing editor as well as addressing the list of requirements or publication in the journal. Please review and adhere to the Journal's requirements for rigour and reproducibility as outlined here: <https://physoc.onlinelibrary.wiley.com/pb-assets/hub-assets/physoc/documents/TJP-Rigour-and-Reproducibility-Requirements-1724673661727.pdf>

Some specific items noted:

Please indicate the diet and water access of the mothers (age and weight if known). State any housing enrichments or conditions specific to the study. State exclusion criteria.

State the distribution of sex of the pups if known, and whether sex was tracted as a cofactor. If not, please indicate.

Please provide RRIDs of antibodies if available. <https://scicrunch.org/resources>

REFEREE COMMENTS

Referee #1:

The authors investigate the morphological and functional effects of sympathetic innervation on cardiomyocytes using an in vitro co-culture system of neonatal rat-derived primary sympathetic neurons (SNs) and human induced pluripotent stem cell--

derived cardiomyocytes (hiPSC-CMs). The authors demonstrate that stimulation of SNs leads to an increased in cytosolic cAMP levels in the co-cultured hiPSC-CMs. This finding is significant as it highlights a direct molecular response of CMs to sympathetic stimulation.

Major Comments

In Fig. 2, the authors should clarify the definition of a "neurocardiac junction" in their co-culture system. Are the authors defining the neurocardiac junction as a direct association between the SN soma and hiPSC-CM, or as a synaptic junction between the axonal terminal of SN and hiPSC-CM? Additionally, how do the authors identify SN soma in the co-culture? Do the authors detect DAPI+ nuclei within the SN soma?

In Fig. 4, understanding the data on the cytosolic cAMP response measured with the FRET sensor is somewhat challenging.

- It would be helpful if the authors could revise Figs. 4A and B by inverting the FRET ratio to CFP/YFP instead of YFP/CFP, so that the ratio increases with rising cytosolic cAMP levels. Additionally, the authors should present multiple representative datasets to support their findings.
- The authors should show representative images of the cytosolic cAMP response in the mono-cultured and co-cultured hiPSC-CMs and indicate the location of their FRET measurements.
- Does isoproterenol, a selective agonist of β -adrenergic receptors, elicit a similar cytosolic cAMP response in the mono-cultured hiPSC-CMs as nicotine does in the co-cultured hiPSC-CMs?
- In Fig. 4A, it appears that cAMP levels gradually increase over time in the monocultured hiPSC-CMs. Does nicotine directly trigger the cytosolic cAMP response in the monocultured hiPSC-CMs, or is the increase intrinsic to the monocultured hiPSC-CMs?
- In Fig. 4B, there is no response to 1 μ M nicotine. But Fig. 4D shows there is some response. Is there a better plot for Fig. 4B?
- The cAMP response data in Figs. S1I and J are unclear. How were the isoproterenol plots generated, and how was the cAMP response normalized?

In Fig. 5, the authors should provide a representative image of the hiPSC-CMs analysed for contraction and calcium levels, clearly indicating the regions where measurements were taken.

In Figs. 6 and 7, it is unclear why the co-cultured hiPSC-CMs did not exhibit a robust increase in contraction and cytosolic calcium in response to nicotine. This result is unexpected, given that nicotine stimulates SNs (Figure 2) and increases the cytosolic cAMP response in the co-cultured hiPSC-CMs (Figure 4).

Minor Comments

- In Materials & Methods, did the authors perfuse the monocultured and co-cultured hiPSC-CMs with buffer during imaging?
- In Materials & Methods, the authors need to provide a brief description of the liquid junction potential and how/why it was needed for their AP recordings.
- In Materials & Methods, the authors need to better describe their cAMP FRET sensor and method. It seems their FRET reaction is measuring a conformational change. When intracellular cAMP increases, it binds to Epac and causes the fluorophores to separate, reducing FRET. Greater cAMP levels result in less FRET (i.e. more CFP and less YFP emission).
- In Materials & Methods, the authors need a description of how the calcium levels were measured.
- In Abstract, please remove the sentence, "Structural proteins and morphology were investigated with confocal microscopy." It is a repeat. Authors already stated, "Structural changes in hiPSC-CMs were analysed by microscopy techniques."

- In Abstract, please remove the word "also". "Furthermore, the bAR response of cocultured hiSPC-CMs was also larger..."
- In Introduction, the authors should provide an additional sentence or reference paper explaining the "fight or flight" response.
- In Introduction, according to the paper by Kowalski et al. SNs from mouse embryos, rather than rat embryos, were used for their co-culture system.
- In Introduction, given that the authors did not measure contractile force directly, the sentence that "SNs promote....improved contractile force.." should be rephrased.
- In Text, the authors state that hiPSC-CMs do not express nAChRs, but no references are provided.
- In Fig. 3, there is no panel "I".
- In Discussion, the authors should discuss the role of cAMP in CM maturation. According to the paper (Giacomelli et al. Cell Stem Cell 2020), increased cAMP enhances maturation of hiPSC-CMs.

Referee #2:

Overall, Mohammadi et al. provide a comprehensive study of the neuro-cardiac junction in non-compartmentalized co-culture. The approach is quite thorough, incorporating intracellular calcium handling, mechanobiology, immunohistostaining, FRET, and patch-clamp techniques. They demonstrate the impact of rat sympathetic neurons on improving the morphology and some key functional properties of hiPSC-CMs, although this was only shown using one hiPSC line. Some conclusions are premature and not well-supported experimentally. This work would be stronger if human neurons were used instead of rat ones, given the unverified neurotransmitter release and heterogeneous hiPSC-CMs.

More experiments should be conducted in a major revision, particularly to confirm protein expression changes. Please see my comments below:

Please explain how the cell density in monoculture was adjusted to be comparable with the co-culture condition. Were there clusters of hiPSC-CMs in monoculture while only single hiPSC-CMs in co-culture? Please clarify.

Please provide further details about the hiPSC line used. What is the gender and passage? Was this line characterized in another study? Please clarify. No other hiPSC line has been tested to reinforce the observations seen with the IMR90 line.

Fig. 2: The microscopic data on the neurocardiac junction with synapse labeling are quite convincing. What neurotransmitters are released to modulate the hiPSC-CMs?

Panel 2D: How was the sarcomere organization measured? Please explain.

Fig. 3: Demonstrating further maturity of dissociated hiPSC-CMs in co-culture using immunohistostaining sounds challenging. Given the morphological variation among hiPSC-CMs, the results may not be pertinent despite a substantial sample size (~180 cells). Protein expression of cardiac markers should be used to prove that the co-culture improves the hiPSC-CM cytoarchitecture.

Therefore, protein expression experiments of some key sarcomeric proteins (actin, myosin, titin, etc.) should be performed

to support that co-culture improves sarcomere organization and changes the morphology of the hiPSC-CMs.

Fig. 4: Monitoring the secreted neurotransmitters is needed to support these data, particularly in the context of adult rat SNs and human fetal cardiomyocytes.

Fig. 5: How does the beat rate compare in mono- and co-culture? Please explain why this parameter is not displayed.

Fig. 8: The electrophysiological data are not convincing. Are these hiPSC-CMs supposed to be ventricular-like? The APD90 does not support this. What about the resting membrane potential, which is unaffected by the co-culture? How do these results reinforce the morphological changes and cAMP production by the co-culture? Please explain.

Supp. Material / panel H: How can the authors expect changes in mitochondrial respiration while the glucose-fed hiPSC-CMs are under the glycolytic pathway? Were fatty acids applied during cardiac differentiation? Please explain.

The discussion section lacks recent work using rat PC12 cells/neurons to modulate SR calcium handling and contraction of hiPSC using NGF induction (please see PMID: 36497024 from 2022).

END OF COMMENTS

EDITOR COMMENTS

Reviewing Editor:

Your paper has been reviewed by two experts in the field. They felt the studies comprised a thorough experimental approach, which has led to the compelling and important finding that neuronal stimulation leads to increased cytosolic cAMP in hiSPC-CM, which enhances their function and maturity. However, they also felt that some of the work was premature in nature, such that conclusions were not well-supported by the experimental results, thus requiring additional experiments and analyses along with a major revision of the paper as outlined in the Comments for the Author. If you are willing to conduct this additional work and make the required revisions, please submit a revised manuscript, along with a point-by-point response to the reviewers' concerns.

Please also see 'Required Items' above.

Senior Editor:

Thank you for submission of your research article to the Journal of Physiology for consideration. The article has been reviewed by experts in the field and found to be potentially acceptable for publication pending a major revision to address all of the concerns raised. It was deemed that additional experiments are needed to address some experiments found to be premature for conclusions raised as outlined in the reviews. Please address all comments from the external referees and reviewing editor as well as addressing the list of requirements or publication in the journal. Please review and adhere to the Journal's requirements for rigour and reproducibility as outlined [here:https://physoc.onlinelibrary.wiley.com/pb-assets/hub-assets/physoc/documents/TJP-Rigour-and-Reproducibility-Requirements-1724673661727.pdf](https://physoc.onlinelibrary.wiley.com/pb-assets/hub-assets/physoc/documents/TJP-Rigour-and-Reproducibility-Requirements-1724673661727.pdf)

We thank the editor for this feedback. We have now completed multiple additional experiments addressing points raised by the reviewers and further revised the text in line with comments raised by reviewers, the reviewing editor and the new data. We have also addressed the list of requirements for publication in the journal. Together our manuscript has been much improved.

Some specific items noted:

Please indicate the diet and water access of the mothers (age and weight if known). State any housing enrichments or conditions specific to the study. State exclusion criteria.

The following text has been added to the methodology:

“The pregnant mothers weighed between 200 and 300 g on the day of mating. They were provided with a standard Sniff diet and had free access to water. Housing enrichment included a tunnel and shredded tissue.”

“Exclusion criteria were applied in cases of unhealthy litters, defined as instances where more than half of the litter was non-viable. No cell isolation was performed under these circumstances.”

State the distribution of sex of the pups if known, and whether sex was tracked as a cofactor. If not, please indicate.

The distribution of the pups' sex is unknown. Verifying the sex of day 1 or 2 pups is extremely difficult and prone to errors. Consequently, sex was not tracked as a cofactor. However, all pups from each litter were used, ensuring that the population of sympathetic neurons utilized in this study was a mix of male and female origins, with an estimated ratio of approximately 50/50.

The following information has been added to the Methods section to clarify this point:

“All the pups from each litter were culled and used in subsequent steps, without distinguishing between males and females.”

Please provide RRIDs of antibodies if available. <https://scicrunch.org/resources>

We have added the RRIDs numbers on the tables for the antibodies registered on the RRID site.

REFEREE COMMENTS

Referee #1:

The authors investigate the morphological and functional effects of sympathetic innervation on cardiomyocytes using an in vitro co-culture system of neonatal rat-derived primary sympathetic neurons (SNs) and human induced pluripotent stem cell-derived cardiomyocytes (hiPSC-CMs). The authors demonstrate that stimulation of SNs leads to an increased in cytosolic cAMP levels in the co-cultured hiPSC-CMs. This finding is significant as it highlights a direct molecular response of CMs to sympathetic stimulation.

We thank the reviewer for recognising the significance of our work.

Major Comments

In Fig. 2, the authors should clarify the definition of a "neurocardiac junction" in their co-culture system. Are the authors defining the neurocardiac junction as a direct association between the SN soma and hiPSC-CM, or as a synaptic junction between the axonal terminal of SN and hiPSC-CM? Additionally, how do the authors identify SN soma in the co-culture? Do the authors detect DAPI+ nuclei within the SN soma?

We thank the reviewers for pointing out that we had not defined the term "neurocardiac junction." We have now included a definition in the text:

“The neurocardiac junction is defined as the area of contact between the SN neurites and the CM surface.”

We specifically define the neurocardiac junction as the association between the synaptic junction from the axonal terminal of the SN and the surface of the hiPSC-CM. The SN soma can be easily identified in immunostainings by DAPI staining due to its larger size compared

to the nuclei of other cell types, as well as the Z position, given that neuron somas in our co-culture conditions are always above the hiPSC-CM layer.

In SICM experiments, somas can also be easily identified in brightfield imaging based on their size and morphology. The longest neurite extending from the soma is typically identified as the axon. However, as we did not use an axonal marker, we prefer to adopt a conservative definition in our study.

We have also modified the title of Figure 2 (Figure 3 in this revised version) to better represent the distinction between the functional analysis of the SNs, and the morphological maps of the NCJ as follow:

“Figure 3. Functional activity of SNs and morphological characteristics of the neurocardiac junction (NCJ).”

In Fig. 4, understanding the data on the cytosolic cAMP response measured with the FRET sensor is somewhat challenging.

- **It would be helpful if the authors could revise Figs. 4A and B by inverting the FRET ratio to CFP/YFP instead of YFP/CFP, so that the ratio increases with rising cytosolic cAMP levels. Additionally, the authors should present multiple representative datasets to support their findings.**

Historically, we tend to express the result of FRET experiments showing YFP/CFG ratio for all the sensors. But some of the sensors increase fluorescent energy transfer, whereas some sensors decrease this transfer upon the binding of cAMP. Indeed, as clearly understood by the reviewer, in the case of Epac-SH74 the FRET diminishes as the cAMP level increases. We are now clearly indicating this in the methods and in the figure legend. We have kept the figures unchanged as it maintains consistency with all our previously published work. (PMID: 37869877; PMID: 32228862; PMID: 24345421; PMID: 29642004; PMID: 30283354; PMID: 20185685)

In the new Figure 7, we present two representative traces for each condition, with data from approximately 60 cells from 5 different batches in monocultures and cocultures. We consider this a high-quality representation of the cell population, and the findings are further supported by statistical analysis.

- **The authors should show representative images of the cytosolic cAMP response in the mono-cultured and co-cultured hiPSC-CMs and indicate the location of their FRET measurements.**

We thank the reviewer for this comment and apologize for not including more detailed information in the methodology section. We have now expanded the methodology section for the FRET measurements to provide a more comprehensive explanation of the process. The FRET measurements were not performed at a specific cellular location; instead, the entire cell was selected as the region of interest (ROI) for analysis, with the background subtracted from an area outside the cell.

As a reference, we took screenshots of the acquisition software as per the figure below (Figure A), where examples from both a monoculture and a coculture dish are presented. Since neurons were not transfected in our conditions, cocultures appear identical to monocultures. Unfortunately, we do not have the means to capture images of the cytosolic

cAMP response, as the software does not capture those images. The cAMP response is instead represented by the provided traces.

We have not included representative images of the cells because we consider that the image quality of a screenshot is inadequate and falls short of the 600-dpi resolution expected for published images.

Figure A. Representative fluorescent images of hiPSC-CM transfected with the Epac-SH74 from monoculture and coculture.

- **Does isoproterenol, a selective agonist of β -adrenergic receptors, elicit a similar cytosolic cAMP response in the mono-cultured hiPSC-CMs as nicotine does in the co-cultured hiPSC-CMs?**

Yes, the average 10 μ M nicotine response in the cocultures corresponds to approximately a 50% increase, which is comparable to the response elicited by 10 nM isoproterenol (ISO) observed in the dose-response curve (Figure 7D). This indicates that nicotine triggers the release of the amount of NE comparable to 10nM isoproterenol. In monocultures, 10 nM ISO also elicits the same 50% increase as observed in cocultures. This indicates that monocultures have the same potential to respond to β -adrenergic receptor stimulation.

- **In Fig. 4A, it appears that cAMP levels gradually increase over time in the monocultured hiPSC-CMs. Does nicotine directly trigger the cytosolic cAMP response in the monocultured hiPSC-CMs, or is the increase intrinsic to the monocultured hiPSC-CMs?**

We apologize for the unclear trace in Figure 4A. The constant "increase" observed in the trace is an artifact caused by a slight drift in the image over time, which may occur in some cells. This drift is not intrinsic to the monoculture hiPSC-CMs; it can happen in either direction and similar traces have been observed in the cocultures.

To provide a better representation of the experiments, we have now included four traces in the new Figure 7. While this artifact is corrected during analysis by adjusting the full trace to the baseline drift, we have chosen to display raw traces to maintain transparency.

Figure B. (New Figure 7 – Panel A) Representative traces of FRET response from monoculture and coculture, to 1 μ M and 10 μ M nicotine, 100 μ M of the PDE inhibitor IBMX, and 10 μ M of the adenylyl cyclase activator NKH, to normalise to the maximum cAMP response. Notice that upon binding to cAMP, the sensor exhibits a decrease in FRET.

- In Fig. 4B, there is no response to 1 μ M nicotine. But Fig. 4D shows there is some response. Is there a better plot for Fig. 4B?

The response to 1 μ M nicotine was small in some cocultures cells. To provide a better representation of the cell population, we are now displaying four traces, including two traces from coculture recordings.

- The cAMP response data in Figs. S1I and J are unclear. How were the isoproterenol plots generated, and how was the cAMP response normalized?

We apologize for the lack of clarity in the explanation of the ISO concentration-response curve. A clearer explanation has now been added to the methodology, as follows:

“For the ISO dose-response curve, cells were stimulated with ISO concentrations of 0.1, 0.3, 1, 3, 10, 30, and 100 nM, followed by IBMX and NKH (as per nicotine data). The data were plotted as a percentage of the cAMP response, normalized to NKH. ISO concentrations were

log-transformed, and the means of each subcolumn were normalized to produce the concentration-response curve.”

We have also modified the figure to represent the standard deviation, rather than the standard error as previously shown, in accordance with the Journal of Physiology guidelines.

In Fig. 5, the authors should provide a representative image of the hiPSC-CMs analysed for contraction and calcium levels, clearly indicating the regions where measurements were taken.

The Cytocypher software uses a semi-automatic procedure that acquires thumbnail images of each cell it investigates. We are showing a screenshot of the software in the figure below (Panel A), with some thumbnail images (Panel B), it can be appreciated that the quality of the images is low. The area under the purple rectangle in panel B is the region of measurement, a fixed area recorded by the high-speed camera at the center of the frame, which cannot be modified by users.

For calcium fluorescence, no images are generated by the Cytocypher. Instead, the whole-cell fluorescence signal is measured using a photomultiplier tube (PMT). The successful loading of Fura-2 AM into the cells is verified by the presence of a ratiometric signal that can be measured with the cell contract. The region of measurement corresponds to the full area detected by the photomultiplier.

To address this point, we have included the following clarification in the Cytocypher methodology:

“Thumbnail images of each investigated cell were acquired using a high-speed MyoCam-S3 digital camera, and whole-cell fluorescence photometry of Fura-2 AM was measured with a photomultiplier tube. The region of measurement was automatically selected by the software as the center of the camera frame, and the full area detected by the photomultiplier was analyzed.”

Figure C. A) Screenshot of the Cytocypher software showing different regions of the dish and the area that is selected prior recording of the cell. B) Representative images of monoculture and coculture hiPSC-CMs. The area of measure is determine by the purple margins.

In Figs. 6 and 7, it is unclear why the co-cultured hiPSC-CMs did not exhibit a robust increase in contraction and cytosolic calcium in response to nicotine. This result is unexpected, given that nicotine stimulates SNs (Figure 2) and increases the cytosolic cAMP response in the co-cultured hiPSC-CMs (Figure 4).

We agree with the reviewer's comment that a robust increase in contraction and cytosolic calcium levels would be expected given the cAMP responses. We were also initially puzzled by the apparent lack of effect. This uncertainty prompted us to conduct experiments with a larger number of batches and cells in this set of experiments to ensure the validity of our findings.

We think this is a result of a technical difficulty linked to the fact that high-affinity calcium dyes have been reported to buffer intracellular calcium and alter their physiological behaviour (Wokosin et al., 2004; Fast, 2005). This effect is evident in our monoculture experiments, where contraction and calcium parameters diminish over time. We are confident that this technical limitation is masking the expected nicotine-induced inotropic effect on innervated cardiomyocytes. The observed effects are likely influenced by the buffering properties of the calcium dyes and the limitations of the technique.

Notably, when the results are plotted as a percentage change before and after nicotine application (Figure D), the effect of nicotine on cocultures becomes clearer. However, we believe it is more appropriate to present the results as shown in the original figures. Representing the data as percentage change could be misleading, potentially overstating the effect observed. Our aim is to present the recorded results accurately without creating an impression of a stronger effect than what is evident.

Figure D. Representation of pixel correlation peak height change as a percentage when normalized to the baseline value on each cell.

We have expanded the following text to clarify this limitation:

“Unfortunately, high-affinity calcium dyes have been reported to buffer intracellular calcium and alter physiological behaviour (Wokosin et al., 2004, Fast, 2005). This limitation is likely exacerbated under the conditions of our recordings—constant pacing at 1 Hz, 37°C, and Fura-2 AM loading—which may cause a progressive decrease in the contraction capacity of

the cells over time. As a result, these effects could be masking the positive inotropic effects that would otherwise be expected in the cocultures.”

Minor Comments

• In Materials & Methods, did the authors perfuse the monocultured and co-cultured hiPSC-CMs with buffer during imaging?

All immunostaining images were obtained from fixed cells without perfusion. We note that even for mitochondrial analysis, despite making use of a mitochondrial dye (mitotracker) that is added to live cells, cells were fixed and immunostained prior to downstream analysis.

For contraction and calcium imaging, cells were live but not perfused because analysis is based on pixel correlation and constant perfusion would introduce motion artifacts leading to and any pixel movement unrelated to cell contraction. However, we note that the chamber and all solutions were maintained at 37°C, and drugs were applied in a single application for each experiment.

SICM and FRET images were also acquired without a perfusion system. These systems operate at room temperature to prevent cell contraction and minimize motion artifacts, eliminating the need for perfusion.

We thank the reviewer for raising this question and apologize for omitting this detail previously. We have now clarified in the methods section that these experiments were performed at room temperature.

• In Materials & Methods, the authors need to provide a brief description of the liquid junction potential and how/why it was needed for their AP recordings.

The liquid junction potential (LJP) occurs when two liquids in contact with each other have different ion concentrations. All electrophysiological patch-clamp experiments require an adjustment for their corresponding LJP.

In our case, we calculated the LJP using LJPcalc, as explained in the methods. Since the LJP was not negligible, we corrected it in all recordings offline using the following formula: $V_m = V_{rec} - LJP$, where V_{rec} is the measured voltage from the scaled output of the amplifier.

To clarify this point, we have included the following text in the methods section:

“The liquid junction potential (LJP) occurs when two liquids in contact with each other have different ion concentrations, and it must be adjusted to accurately measure the cell voltage.”

• In Materials & Methods, the authors need to better describe their cAMP FRET sensor and method. It seems their FRET reaction is measuring a conformational change. When intracellular cAMP increases, it binds to Epac and causes the fluorophores to separate, reducing FRET. Greater cAMP levels result in less FRET (i.e. more CFP and less YFP emission).

We have expanded the methods section to provide additional details:

“To measure cytoplasmic cAMP levels, the m-Turquoise based TEpacVV cAMP FRET sensor Epac-SH74 sensor (Klarenbeek et al., 2011), generated in our laboratory, consists of a mutated full-length EPAC1 protein serving as the sensing element. This protein is flanked by mTurquoise (excitation: 434 nm, emission: 474 nm) as the donor fluorophore and a Venus dimer (excitation: 515 nm, emission: 528 nm) as the acceptor. Upon binding to cAMP, the sensor exhibits a decrease in FRET.”

Additionally, we have included the formula used to calculate the FRET ratio shifts.

- **In Materials & Methods, the authors need a description of how the calcium levels were measured.**

The following text has been added to describe the measurement of calcium levels:

“Fura-2 AM binds to calcium ions in the cytosol, resulting in a shift in its fluorescence excitation spectrum. The Cytocypher system excites the dye at two required wavelengths (340 nm and 380 nm). The fluorescence intensity ratio (340/380 nm) is then calculated for each cell, providing a ratio proportional to the intracellular calcium concentration.”

- **In Abstract, please remove the sentence, "Structural proteins and morphology were investigated with confocal microscopy." It is a repeat. Authors already stated, "Structural changes in hiPSC-CMs were analysed by microscopy techniques."**

Thanks for noticing this error. It has been amended.

- **In Abstract, please remove the word "also". "Furthermore, the bAR response of cocultured hiSPC-CMs was also larger..."**

It has been removed.

- **In Introduction, the authors should provide an additional sentence or reference paper explaining the "fight or flight" response.**

Additional information explaining the "fight or flight" response has been added to the introduction as requested:

“The sympathetic branch of the ANS is particularly important as it regulates the body’s “fight or flight” response, a physiological reaction triggered during acute stress. This response is mediated by the ANS, particularly the sympathetic system, through paracrine secretion of neurotransmitters—mainly norepinephrine (NE)—across the synaptic interface, termed the “neuro-cardiac junction” (NCJ)” (Zaglia and Mongillo, 2017), which prepares the organism for rapid action.”

- **In Introduction, according to the paper by Kowalski et al. SNs from mouse embryos, rather than rat embryos, were used for their co-culture system.**

We apologize for the mistake. It has been changed.

- **In Introduction, given that the authors did not measure contractile force directly, the sentence that "SNs promote....improved contractile force.." should be rephrased.**

The reviewer is correct. It has now been changed.

- **In Text, the authors state that hiPSC-CMs do not express nAChRs, but no references are provided.**

We apologize for this overstatement. Nicotinic receptors are lowly expressed in cardiomyocytes. We searched for their expression levels in RNAseq datasets e.g. in the heart cell atlas (<https://www.heartcellatlas.org/v2/global/>) for the expression of their genes (CHRNA1-10). For iPSC-cardiomyocytes we used as reference our own published dataset (GSE203375, PMID 37159667) that confirmed their low expression levels. However, we were unable to find any studies reporting the presence of nAChRs in hiPSC-CMs, or identify any study that explicitly reported their absence. As a result, we have removed the sentence from the text.

- **In Fig. 3, there is no panel "I".**

Thanks for spotting the error. Mention to panel "I" has been removed.

- **In Discussion, the authors should discuss the role of cAMP in CM maturation. According to the paper (Giacomelli et al. Cell Stem Cell 2020), increased cAMP enhances maturation of hiPSC-CMs.**

We thank the reviewer for this suggestion. Indeed, Giacomelli et al. (2020) highlighted the role of cAMP in promoting cardiomyocyte maturation. Similar to their observations in cultures treated with dbcAMP, we also observe an increase in AP amplitude and depolarization velocity in our co-cultures. However, we do not observe other changes, such as in the sarcomere alignment index or membrane potential alterations, suggesting that different mechanisms may be at play.

We have revised the discussion to emphasize that the interaction of cardiomyocytes with sympathetic neurons leads to a stronger cAMP response within cardiomyocytes. This observation suggests increased cAMP levels within the cells. However, our FRET measurements cannot report baseline cAMP levels, so we cannot conclusively confirm this.

New text has been added to the discussion:

"An increase in cAMP levels has also been linked to enhanced maturation of hiPSC-CMs, including improved sarcomere organization and greater amplitude and upstroke velocity of the action potential (Giacomelli et al., 2020). Although FRET cannot directly report baseline cAMP levels in our case, it is plausible that SNs may alter baseline cAMP levels and further promote maturation."

Referee #2:

Overall, Mohammadi et al. provide a comprehensive study of the neuro-cardiac junction in non-compartmentalized co-culture. The approach is quite thorough, incorporating intracellular calcium handling, mechanobiology, immunohistostaining, FRET, and patch-clamp techniques. They demonstrate the impact of rat sympathetic neurons on improving the morphology and some key functional properties of hiPSC-CMs, although this was only shown using one hiPSC line. Some conclusions are premature and not well-supported experimentally. This work would be stronger if

human neurons were used instead of rat ones, given the unverified neurotransmitter release and heterogeneous hiPSC-CMs. More experiments should be conducted in a major revision, particularly to confirm protein expression changes. Please see my comments below:

We are grateful for the suggestion to include a second cell line. In this resubmission, we have included results from hESC-CMs in key experiments. These include analyses of cell morphology, sarcomere length and organization, metabolism, contraction, and calcium transients. We believe that the addition of a second cell line, which demonstrates the same effects observed in hiPSC-CMs, significantly strengthens the conclusions of our work.

Please explain how the cell density in monoculture was adjusted to be comparable with the co-culture condition. Were there clusters of hiPSC-CMs in monoculture while only single hiPSC-CMs in co-culture? Please clarify.

We apologize for not including that monocultures were seeded in parallel to the cocultures, under the same density and conditions, except for the addition of neurons. As mentioned, in cocultures, sympathetic neurons (SNs) were mixed into the cell suspension medium at a ratio of 1:20 (one SN per cardiomyocyte) and seeded onto glass coverslips. The same protocol was used for monocultures, but without the addition of SNs before seeding. This information has now been added to the methods section:

“Batch-matched monocultures were seeded in parallel to cocultures following the same protocol and conditions, but without adding the SNs.”

A mix of clusters and single cells were observed in the different experiments, depending on the batch of cells, but the proportion of clusters and single cells was similar between monocultures and cocultures.

Please provide further details about the hiPSC line used. What is the gender and passage? Was this line characterized in another study? Please clarify. No other hiPSC line has been tested to reinforce the observations seen with the IMR90 line.

To strengthen the conclusions drawn from the IMR90 cell line, we have included a new set of experiments using the H9 cell line in this updated version.

These cell lines have been utilized by other research groups, including ours, as referenced in Hasan et al., 2020 (PMID: 33053822), Faleeva et al., 2022, (PMID: 36291124), Dark et al., 2023 (PMID: 33053822),

The efficiency of differentiation was easily assessed due to the distinct nature of the differentiated cell type, with cardiomyocytes (CMs) forming a monolayer and exhibiting beating behaviour in culture. These observations are also backed up by data in Figure 1 confirming that a high percentage of cardiomyocytes was obtained across various differentiation batches

The following details have been incorporated into the new methods section: Origin of Human Pluripotent Stem Cell Lines:

“The hPSC line IMR90 is an induced pluripotent stem cell line derived from human female fetal lung fibroblasts and was purchased from WiCell Technologies (Madison, WI, USA). Similarly, the hPSC line H9 (WA09, karyotype: 46, female), a human embryonic stem cell line, was also obtained from WiCell Technologies. Since two hPSCs were used, these will be

referred to as hiPSCs for the IMR90 line and hESCs for the H9 line. The hPSC lines were maintained in feeder-free culture conditions on Corning growth factor reduced Matrigel membrane matrix (GFR Matrigel, Corning) for hESCs or geltrex (Thermo Fisher) for hiPSCs, in mTeSR1 (STEMCELL Technologies) for hESCs or Essential-8 (E8) (Gibco, A1517001) for hiPSCs maintenance medium. Cells were passaged every 4–5 days as aggregates using Gibco Versene solution (Thermo Fisher Scientific) and cell scrapers, at a split ratio of 1:13 (hiPSCs) or 1:8-1:10 (hESCs). All experiments were performed within 20 passages from thawing. All experiments with hESCs were approved by the UK Stem Cell Bank steering committee.”

Fig. 2: The microscopic data on the neurocardiac junction with synapse labeling are quite convincing. What neurotransmitters are released to modulate the hiPSC-CMs?

Based on our results, the fact they were neurons isolated from sympathetic ganglia, the positive TH staining, and the cAMP activation on the CMs comparable to 10nM ISO, together with the existing literature, we are confident that the primary modulator of our hiPSC-CMs is norepinephrine (NE), the main neurotransmitter released by sympathetic neurons upon stimulation. Therefore, the focus of this work is on the downstream signalling of β -adrenergic receptors (BARs). However, sympathetic neurons can also release co-transmitters such as Neuropeptide Y (NPY) and ATP, although significant co-transmitter release typically occurs only during high-level neuronal stimulation. ATP is rapidly metabolized upon release and is unlikely to play a role under our experimental conditions. On the other hand, NPY cannot be entirely ruled out. Recent work (<https://doi.org/10.1093/eurheartj/ehae666.3774>) describes that hiPSC-CMs express the NPY receptors Y1 and Y5, both at the mRNA and protein levels. However, the use of NPY agonists led to a slowing of repolarization and arrhythmogenic behaviours in the cells, as also reported by other authors (PMID: 38744275). Indeed, NPY reduces cAMP levels via adenylyl cyclase inhibition, which is the opposite of the effect observed in our work after nicotine application. This suggests that any potential effect of NPY is likely compensated by the higher release of NE, as described in the literature.

Panel 2D: How was the sarcomere organization measured? Please explain.

In our previous manuscript version the sarcomere analysis was described within the confocal analysis section. We have now separated the sarcomere analysis and extended the method details:

“Sarcomere analysis

For the analysis of sarcomere length, organization, and alignment, the software tool for automatic quantification (SOTA) was used (Stein et al., 2022). An automatic Python application base on a sarcomere analysis MATLAB script. In brief, single cell images were cut from the original raw confocal images for the α -actinin staining. Without any modification, images were uploaded to the SOTA software and automatically analysed. In contrast to other approaches, the method does not require regions of interest (ROIs) to be pre-defined or selected, or images of fluorescent sarcomeres to be aligned manually. SotaTool detects the optimal angle automatically, reducing analysis time and selection bias.”

Specifically for the sarcomere organization the software uses a two pre-processing steps (background, subtraction, and segmentation), the gray-level co-occurrence matrix (GLCM) is calculated, and the raw data is saved per image as a comma delimited file (.csv). The curve with the highest peak is isolated from all the generated curves and the sarcomere

organization score is calculated as the difference between the height of the peak and its preceding minimum. At the offset distance of the maximum peak, the variance between all curves is taken as an alignment index. The sarcomere length is calculated from the maximum peak after interpolating with a bicubic spline, for sub-pixel accuracy.

The paper (PMID: 35789134) includes a step-by-step guide to upload the figures into the Python application to obtain the sarcomere parameters automatically from the original confocal images of α -actinin.

Fig. 3: Demonstrating further maturity of dissociated hiPSC-CMs in co-culture using immunohistostaining sounds challenging. Given the morphological variation among hiPSC-CMs, the results may not be pertinent despite a substantial sample size (~180 cells). Protein expression of cardiac markers should be used to prove that the co-culture improves the hiPSC-CM cytoarchitecture. Therefore, protein expression experiments of some key sarcomeric proteins (actin, myosin, titin, etc.) should be performed to support that co-culture improves sarcomere organization and changes the morphology of the hiPSC-CMs.

We apologize if our previous version of the manuscript led to any misunderstandings regarding the conclusions of our work. As mentioned, all the changes we observed were morphological or functional in nature but not related to alterations in protein expression. Finding a reliable protocol to separate the protein fraction of the hiPSC-CMs from the sympathetic neurons (SNs) in the coculture conditions has been elusive so far. However, we are still testing different approaches and hope to include these results in future work. Unfortunately, we have not yet succeeded in performing protein expression analysis.

Nevertheless, we have leveraged our mixed in vitro model (rat and human) to perform gene expression analysis of structural related genes. By designing human-specific primers, we were able to separate the gene expression of the hiPSC-CMs. These results have been included in the new Figure 5, and expanded on these results within the methods, results, and discussion sections of the manuscript. We did not find any significant changes between monocultures and cocultures of hiPSC-CMs, which strongly suggests that the functional changes observed in our 10-day cocultures are not due to changes in gene expression, but are most likely caused by the reorganization of key proteins, such as sarcomeric proteins, rather than an increase in protein levels. This reasoning has been included in the Discussion as follows:

“We did not observe any mRNA expression changes for the proteins tested in this study (Fig. 5). Although this does not rule out a possible increase in protein levels, our findings on sarcomere organization and functional changes suggest that the improved cytoarchitecture observed in the 10-day cocultures is a consequence of an enhanced cellular function induced by the SNs.”

Figure E. (New Figure 6, Panel E) RT-qPCR confirmed a lack of significant change in the expression of key structural genes involved in hiPSC-CM contraction. Data presented as mean with SD, Mann-Whitney test, $N \geq 6$. N = biological replicate.

Fig. 4: Monitoring the secreted neurotransmitters is needed to support these data, particularly in the context of adult rat SNs and human fetal cardiomyocytes.

Unfortunately, we haven't succeeded in this endeavour. We faced a technical limitation, we don't have the means to monitor the neurotransmitter release on the neuro-cardiac junction space, to accurately measure how much NE is release on the synaptic cleft space between the neuron and the cardiomyocyte, which will undoubtedly provide a breakthrough in the field.

We believe the reviewer is requesting an ELISA analysis of neurotransmitter release in the culture medium following nicotine application. Our model of sympathetic neurons is well-characterized in the literature, with a pure sympathetic neuron phenotype, where all neurons are positive for tyrosine hydroxylase, and thus NE would be the neurotransmitter to detect.

However, we are doubtful about the feasibility of this approach in our case. To our knowledge, NE has not been successfully detected in co-culture conditions by other groups. The only successful detection of neurotransmitters in similar conditions was by Bernardin et al. (2022; PMID: 36497024), who detected acetylcholine from co-cultures, but they were unable to measure catecholamines using ELISA kits. Winbo et al. (2020; PMID: 32822546) detected NE in the culture medium, but only from 75-day-old hiPSC-derived SN monocultures after a high dose of KCl, which is quite different from our 10-day-old co-cultures with a low proportion of neurons (One neurons per twenty cardiomyocytes plated per dish).

It has been described that the synaptic cleft between neurons and cardiomyocytes is a relatively narrow intercellular space (80–100 nm wide), in which NE is preferentially discharged due to the polarization of the neuronal active zone. This allows for a high concentration of NE to be achieved upon the release of relatively few molecules, thus efficiently activating cardiac β -adrenergic receptors (β ARs) (PMID: 28240352). Therefore, even if low doses of NE were detected in the culture medium (assuming NE escapes the synaptic cleft without being degraded or reabsorbed by the sympathetic neurons), it would not be indicative of the effect that sympathetic neurons could have on cardiomyocytes.

For example, Dokshokova et al. (2022; PMID: 3541313) detected nerve growth factor (NGF) in the coculture medium of sympathetic neurons with cardiomyocytes, but the concentration was 1,000-fold lower than the minimal concentration required for neuronal survival. This contradicts their observation that the neurons survived due to the NGF provided by the cardiomyocytes in culture, without any addition of external NGF. This suggests that the neuro-cardiac junction is an isolated microenvironment, protected from diffusion, and characterized by a high NGF concentration.

We believe that a technical breakthrough is required to measure real-time release of NE from the synaptic cleft to fully support our experiments. Measuring NE release in the culture medium would not strengthen our results, regardless of whether the outcome is positive or negative.

We have included the following text in the limitation section to address this problem:

“Another limitation is that we have not monitored neurotransmitter release within the neuro-cardiac junction space. It has been described that the synaptic cleft between neurons and cardiomyocytes is a relatively narrow intercellular space (80–100 nm wide) where norepinephrine (NE) is preferentially released due to the polarization of the neuronal active zone (Zaglia and Mongillo, 2017). This structural arrangement allows for a high local concentration of NE upon the release of relatively few molecules, efficiently activating cardiac β -adrenergic receptors (β ARs). Consequently, even if low NE levels are detected in the culture medium, as demonstrated by Bernardin et al. (2022) for acetylcholine in cocultures, this would not accurately reflect the localized effects of sympathetic neurons on cardiomyocytes. To fully validate our experiments, a technical breakthrough is required to enable real-time measurement of NE release within the synaptic cleft.”

Fig. 5: How does the beat rate compare in mono- and co-culture? Please explain why this parameter is not displayed.

We thank the reviewer for this comment and have now included these results in the revised version of our manuscript. We measured the baseline beats per minute (bpm) in hiPSC-CMs monocultures and cocultures. As shown in the figure below, there is no significant difference in the beat rate of monoculture and cocultures. We observed however a high variability

between batches, which is in keeping with the fact the standard differentiation protocol used generates a variety of cardiomyocytes with varying percentages of atrial, nodal or ventricular cardiomyocytes. This prompted us to focus on pacing experiments rather than spontaneous beating, as calcium transient parameters (e.g., peak and velocities) and, consequently, contraction, are directly linked to the beating rate. Since we did not continue experiments on spontaneous beating, we initially did not consider these results highly relevant. However, in response to the reviewer's question, we have included these results to provide more details about our CMs' characteristics.

We repeated these measurements on the H9 cell line, albeit only in four batches. Interestingly, the variability in this cell line appears reduced. Nonetheless, we performed pacing experiments for comparison with the IMR90 cell line.

This information has been added to the manuscript as follows:

"First, we measured the beats per minute (bpm) in monocultures and cocultures to study whether SNs affected the spontaneous beating of CMs. hiPSC-CMs monocultures showed an average of 45 bpm (SD = 15.17), compared to 48 bpm (SD = 25.09) for cocultures (measured from six batches). Similarly, hESC-CMs showed 38 bpm (SD = 12.92) in monocultures and 42 bpm (SD = 10.92) in cocultures (measured from four batches). The slight increase observed in cocultures was not statistically significant in either case. Given the relatively high variability in bpm between batches and the influence of the beating rate on calcium transient parameters, all subsequent experiments were conducted under electrical pacing conditions."

hiPSC-CMs beats per minute

hESC-CMs beats per minute

Figure F. Measure of spontaneous beating rate of CMs. 10-25 areas were measured and averaged to obtain one value per batch. Data presented as mean with SD, Mann-Whitney test. 6 batches for hiPSC-CMs, 4 batches for hESC-CMs.

Fig. 8: The electrophysiological data are not convincing. Are these hiPSC-CMs

supposed to be ventricular-like? The APD90 does not support this. What about the resting membrane potential, which is unaffected by the co-culture? How do these results reinforce the morphological changes and cAMP production by the co-culture? Please explain.

We thank the reviewer for their question regarding the ventricular characteristics of our iPSC-derived cardiomyocytes. This prompted us to further characterize these cardiomyocytes based on the APD90/APD50 ratio of the action potential, as described in previous literature (PMID: 21367833). According to this criterion, cardiomyocyte subtypes are defined as follows: APD90/APD50 ≤ 1.4 indicates ventricular-like cardiomyocytes, APD90/APD50 ≥ 1.7 indicates atrial-like cardiomyocytes, and a ratio between 1.4 and 1.7 indicates nodal-like cardiomyocytes.

Our analysis revealed that 44% of the monocultures and 48.15% of the cocultures are ventricular-like, suggesting a mixed population of cardiomyocyte subtypes. These findings align with previous reports using standard iPSC-derived cardiomyocyte differentiation protocols (PMID: 37159667). For further clarity, we have included a table summarizing the proportions of each subtype here:

	Monoculture	Co-culture
Ventricular-like	44%	48.15%
Atrial-like	24%	25.93%
Nodal-like	32%	25.92

The significantly higher action potential (AP) amplitude and faster depolarization rate observed in our study may suggest an increase in the expression of ion channels (or their adult isoforms) involved in the depolarization phase of the AP in iPSC-derived cardiomyocytes cocultured with sympathetic neurons. Similar changes in AP amplitude and depolarization rate have been previously reported in cocultures of iPSC-derived cardiomyocytes with cardiac fibroblasts or in cardiac microtissues (PMID: 32459996). Additionally, an increased depolarization rate has been correlated with increased expression of the adult isoform of the sodium channel SCN5A (PMID: 353940109).

We found that coculturing cardiomyocytes with sympathetic neurons did not affect the resting membrane potential, suggesting that the neurons facilitate only certain aspects of cardiomyocyte maturation. Other mechanisms such as metabolic maturation (PMID: 32697997), or cell types, such as mural cells, have been proposed to enhance other aspects of maturation, including resting membrane potential regulation (PMID: 32459996).

Our electrophysiological results add valuable information how SNs contribute to cardiomyocyte development. While morphological changes, calcium transients, beating dynamics, and cAMP handling revealed improvements in sarcomeric structure, calcium, and cAMP regulation, the electrophysiological data specifically suggest enhanced maturity in ion channel function related to the depolarization phase (likely involving sodium and potassium channels).

To our knowledge, this is the first study to demonstrate that sympathetic neurons can influence the electrophysiological properties of iPSC-derived cardiomyocytes. We aim to investigate the role of ion channels involved in the depolarization phase of action potentials in future projects.

We have included the following text in the Result and in the Discussion sections:

"We also characterized the proportion of ventricular, atrial, and nodal-like hiPSC-CMs based on their APD90/APD50 ratio (Matsa et al., 2011). The analysis revealed that in monocultures, the distribution was 44% ventricular-like, 24% atrial-like, and 32% nodal-like hiPSC-CMs. Similarly, in cocultures, the distribution was 48% ventricular-like, 26% atrial-like, and 26% nodal-like hiPSC-CMs. These findings align with previous reports utilizing standard iPSC-derived cardiomyocyte differentiation protocols (Dark et al., 2023), and suggest that SNs had minimal impact on the population distribution of hiPSC-CMs."

"Similar changes in AP amplitude and depolarization rate have been previously reported in cocultures of hiPSC-CMs with cardiac fibroblasts (Giacomelli et al., 2020). Our study demonstrates that SNs can positively influence the electrophysiological properties of hiPSC-CMs."

Supp. Material / panel H: How can the authors expect changes in mitochondrial respiration while the glucose-fed hiPSC-CMs are under the glycolytic pathway? Were fatty acids applied during cardiac differentiation? Please explain.

Cells can metabolize using the OXPHOS pathway even without fatty acid supplementation. For example, in Dark et al. (2023; PMID: 37159667), cells were grown in the same basal medium without fatty acid supplementation. Despite this, the study demonstrated that cells differentiated in different ways exhibited changes in respiration. Therefore, it was reasonable to enquire whether co-culture with neurons induced metabolic changes.

Moreover, it is important to note that changes in cardiomyocyte metabolism has often been correlated to changes in sarcomere organisation and function (e.g. PMID: 32697997), thus this was an important question to address.

The discussion section lacks recent work using rat PC12 cells/neurons to modulate SR calcium handling and contraction of hiPSC using NGF induction (please see PMID: 36497024 from 2022).

We have revised the discussion to include new references including the one mentioned by the reviewer. Specifically, the following text has been added to the discussion:

"It has already previously shown that neurons can modulate sarcoplasmic reticulum calcium handling and contraction in hPSC-CMs (Bernardin et al., 2022). Specifically, the presence of PC12 neurons was found to increase calcium release velocity at baseline"

Dear Dr Sanchez-Alonso,

Re: JP-RP-2024-287569R1 "Sympathetic Neurons can modify the intrinsic structural and functional properties of hPSC-CMs" by Neda Mohammadi, Laura Fedele, Poornaa Chakravarthy, Vladislav Leonov, Lorenza Tsansizi, Hui Gu, Sama Seyedmousavi, Marie-Victoire Cosson, Andreia Sofia Bernardo, Julia Gorelik, and Jose L. Sanchez-Alonso

Thank you for submitting your manuscript to The Journal of Physiology. It has been assessed by a Reviewing Editor and by 2 expert referees and we are pleased to tell you that it is acceptable for publication following satisfactory minor revision.

Please address all the points raised and incorporate all requested revisions or explain in your Response to Referees why a change has not been made. We hope you will find the comments helpful and that you will be able to return your revised manuscript within 2 weeks. If you require longer than this, please contact journal staff: jp@physoc.org.

REVISION CHECKLIST:

We look forward to receiving your revised submission.

Yours sincerely,

Harold Schultz
Senior Editor
The Journal of Physiology

REQUIRED ITEMS

- Research must comply with The Journal's policies regarding animal experiments (<https://physoc.onlinelibrary.wiley.com/hub/animal-experiments>) and adherence to these policies must be stated in the manuscript.

Authors should confirm in their Methods section that their experiments were carried out according to the guidelines laid down by their institution's animal welfare committee, including an ethics approval reference number. The Methods section must contain a statement about access to food, water and housing, details of the anaesthetic regime: anaesthetic used, dose and route of administration, and method of killing the experimental animals.

- Please include an Abstract Figure file, as well as the Figure Legend text within the main article file. Currently, we are missing the abstract figure legend.

EDITOR COMMENTS

Reviewing Editor:

The authors have addressed the reviewers' concerns, but to comply with the journal's policies, need to:

- Include a legend for the abstract figure.
- State the method for euthanising ('culling') the rat pups.

Senior Editor:

Thank you for submission of your research article to the Journal of Physiology for consideration. The article has been reviewed by the original referees and found to be acceptable for publication pending a minor revision to address a couple of concerns raised by the reviewing editor. Please address these comments. The authors are commended on the quality of the figures and legends.

REFEREE COMMENTS

Referee #1:

The inclusion of additional experiments has significantly improved the paper, and the authors have addressed most of this reviewer's comments. It seems that the question of how activated sympathetic neurons by nicotine influence cardiomyocyte maturation still needs to be explored further in the future. Overall, I believe the revised version should be considered for publication.

Referee #2:

The authors have adequately addressed my concerns. The manuscript has been enhanced with additional experiments and explanations. I have no further comments.

END OF COMMENTS

EDITOR COMMENTS

Reviewing Editor:

The authors have addressed the reviewers' concerns, but to comply with the journal's policies, need to:

- **Include a legend for the abstract figure.**

The legend for the abstract figure had been include in the Figure legend section:

Abstract Figure – Schematic representation of the key differences between innervated and non-innervated hiPSC-CMs. Innervated hiPSC-CMs (right) exhibit several improved characteristics compared to non-innervated hiPSC-CMs (left). These include enhanced sarcomere organisation, a stronger cAMP response under nicotine stimulation, increased contractility with elevated calcium transients, and a faster depolarization rate of the action potential.

- **State the method for euthanising ('culling') the rat pups.**

“were culled” had been replace by: “were sacrificed by cervical dislocation, followed by decapitation as a confirmation method by trained staff” in the Method section

We are grateful to the senior editor and the reviewers for their work and for recommending the article for publication.

Dear Dr Sanchez-Alonso,

Re: JP-RP-2025-287569R2 "Sympathetic Neurons can modify the intrinsic structural and functional properties of hPSC-CMs" by Neda Mohammadi, Laura Fedele, Poornaa Chakravarthy, Vladislav Leonov, Lorenza Tsansizi, Hui Gu, Sama Seyedmousavi, Marie-Victoire Cosson, Andreia Sofia Bernardo, Julia Gorelik, and Jose L. Sanchez-Alonso

We are pleased to tell you that your paper has been accepted for publication in The Journal of Physiology.

Yours sincerely,

Harold Schultz
Senior Editor
The Journal of Physiology

If you would like to receive our 'Research Roundup', a monthly newsletter highlighting the cutting-edge research published in The Physiological Society's family of journals (The Journal of Physiology, Experimental Physiology, Physiological Reports, The Journal of Nutritional Physiology and The Journal of Precision Medicine: Health and Disease), please click this link, fill in your name and email address and select 'Research Roundup':
<https://www.physoc.org/journals-and-media/membernews>

- **TRANSPARENT PEER REVIEW POLICY:** To improve the transparency of its peer review process, The Journal of Physiology publishes online as supporting information the peer review history of all articles accepted for publication. Readers will have access to decision letters, including Editors' comments and referee reports, for each version of the manuscript as well as any author responses to peer review comments. Referees can decide whether or not they wish to be named on the peer review history document.
- You can help your research get the attention it deserves! Check out Wiley's free Promotion Guide for best-practice recommendations for promoting your work at: www.wileyauthors.com/eeo/guide. You can learn more about Wiley Editing Services which offers professional video, design, and writing services to create shareable video abstracts, infographics, conference posters, lay summaries, and research news stories for your research at: www.wileyauthors.com/eeo/promotion.
- **IMPORTANT NOTICE ABOUT OPEN ACCESS:** To assist authors whose funding agencies mandate public access to published research findings sooner than 12 months after publication, The Journal of Physiology allows authors to pay an Open Access (OA) fee to have their papers made freely available immediately on publication.

EDITOR COMMENTS

Reviewing Editor:

This paper is now ready for full acceptance.

Senior Editor:

The editors wish to thank the authors for these final adjustments to the manuscript. The article is now accepted for publication. Congratulations for an interesting and insightful study. Please consider the Journal of Physiology for your future studies.